# AniSDF: Fused-Granularity Neural Surfaces with Anisotropic Encoding for High-Fidelity 3D Reconstruction

**Jingnan Gao** **Zhuo Chen** **Xiaokang Yang** **Yichao Yan**[†]

MoE Key Lab of Artificial Intelligence, AI Institute, Shanghai Jiao Tong University.

**https://g-1nonly.github.io/AniSDF_Website/**

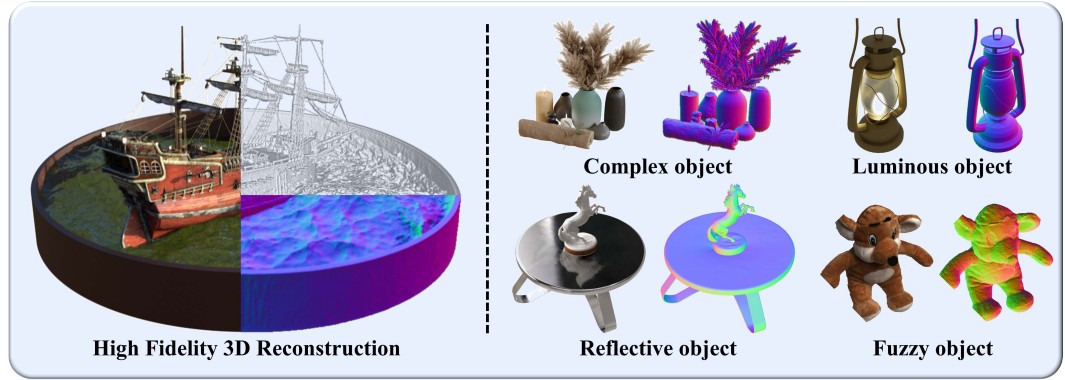

Figure 1: The left part demonstrates the ability of **AniSDF** to produce accurate geometry and high-quality rendering results. The right part presents its capability to handle various scenes including complex object, luminous object, highly reflective object, and fuzzy object.

## ABSTRACT

Neural radiance fields have recently revolutionized novel-view synthesis and achieved high-fidelity renderings. However, these methods sacrifice the geometry for the rendering quality, limiting their further applications including re-lighting and deformation. How to synthesize photo-realistic rendering while reconstructing accurate geometry remains an unsolved problem. In this work, we present AniSDF, a novel approach that learns fused-granularity neural surfaces with physics-based encoding for high-fidelity 3D reconstruction. Different from previous neural surfaces, our fused-granularity geometry structure balances the overall structures and fine geometric details, producing accurate geometry reconstruction. To disambiguate geometry from reflective appearance, we introduce blended radiance fields to model diffuse and specularity following the anisotropic spherical Gaussian encoding, a physics-based rendering pipeline. With these designs, AniSDF can reconstruct objects with complex structures and produce high-quality renderings. Furthermore, our method is a unified model that does not require complex hyperparameter tuning for specific objects. Extensive experiments demonstrate that our method boosts the quality of SDF-based methods by a great scale in both geometry reconstruction and novel-view synthesis.

## 1 INTRODUCTION

Achieving high-quality novel view synthesis and accurate geometry reconstruction are essential long-term goals in the fields of computer graphics and vision. Recently, neural radiance fields (NeRF) Mildenhall et al. (2020) and 3D Gaussian Splatting (3DGS) Kerbl et al. (2023) have achieved photo-realistic rendering results. However, they fail to accurately represent surfaces due to insufficient surface constraints. While these methods trade off geometry accuracy for high-quality render-

---

† : Corresponding author.

ing, accurate geometries are essential to downstream applications such as relighting, PBR synthesis, and deformation. To extract better surfaces while maintaining the appearance quality, several methods Tang et al. (2023b); Rakotosaona et al. (2023) utilize a two-step framework to reconstruct surfaces. However, due to the inevitable loss during the two-step optimization, they fall short in reconstructing high-quality geometric details.

From the perspective of accurate geometry, neural SDF methods Wang et al. (2021a); Yariv et al. (2021); Fu et al. (2022); Yariv et al. (2023); Ge et al. (2023); Li et al. (2023); Wang et al. (2022); Rosu & Behnke (2023); Wang et al. (2023b) emerges to be a possible solution. These methods usually rely on a geometry network to capture the geometric information and an appearance network for rendering. However, appearance learning and geometry learning interact with each other. Specifically, the inability to represent certain appearances will affect the learning process of the corresponding geometry, while the failure to reconstruct accurate geometry in turn affects the optimization of the appearance network. Thus, reconstructing accurate geometry without compromising the rendering quality is a crucial problem for SDF-based methods. To address this issue, some methods Li et al. (2023); Wang et al. (2022; 2023c) adopt a coarse-to-fine training strategy, while other methods Ge et al. (2023); Wang et al. (2023b); Yariv et al. (2023) apply reparametrization techniques or use basic functions Fridovich-Keil et al. (2022); Yu et al. (2021a) to improve the appearance network. However, the trade-off between geometry and appearance remains a problem. The essential challenges for SDF-based methods are (1) modeling fine geometric details and (2) disambiguating geometry from complex appearances such as reflective surfaces.

To address these challenges, our motivations are twofold. First, a fine-detailed geometry highly increases the quality of rendering results. Second, the disambiguation of reflective appearance can significantly reduce the difficulty of learning accurate geometry. We then design our framework from two perspectives. To get detailed **geometry**, instead of using a sequential coarse-to-fine training strategy, we design a parallel structure to learn a fused-granularity neural surface that makes the most of both low-resolution hash grids and high-resolution hash grids. To further disambiguate geometry from **appearance**, we design a blended radiance field to model the diffuse and specularity respectively. We also introduce Anisotropic Spherical Gaussians (ASG) to better model the specular components. By following the physical rendering pipeline, these two networks complement each other and help the model strike a balance between reflective and non-reflective surfaces. We further blend these two radiance fields using a learned weight field, enabling the model to learn scenes including semi-transparent and luminous surfaces. The rendering quality is then improved by a great scale and surpasses both NeRF Mildenhall et al. (2020) and 3DGS Kerbl et al. (2023) and their recent variants.

Overall, we claim the contributions of our paper:

1. We design a unified SDF-based architecture that the geometry network and the appearance network complement each other, producing high-fidelity 3D reconstructions.

2. We present a fused-granularity neural surface to balance the overall structures and fine details.

3. We introduce blended radiance fields with a physics-based rendering via Anisotropic Spherical Gaussian encoding, successfully disambiguating the reflective appearance.

4. Our method boosts the quality of SDF-based methods by a great scale in both geometry reconstruction and novel-view synthesis tasks.

## 2 RELATED WORKS

### 2.1 NOVEL VIEW SYNTHESIS

Neural implicit representations Mildenhall et al. (2020); Lombardi et al. (2019); Loubet et al. (2019); Luan et al. (2021); Lyu et al. (2020); Niemeyer & Geiger (2021); Niemeyer et al. (2020); Pumarola et al. (2021); Yu et al. (2021b); Srinivasan et al. (2021); Barron et al. (2023); Laine et al. (2020); Munkberg et al. (2022) have gained popularity in novel view synthesis. Neural Radiance Field (NeRF) and its follow-up approaches Martin-Brualla et al. (2021); Mildenhall et al. (2020); Park et al. (2021); Zhang et al. (2020); Wang et al. (2021b); Reiser et al. (2023); Fridovich-Keil et al. (2022); Hu et al. (2023); Chen et al. (2022; 2023b); Zhang et al. (2023); Shu et al. (2023); Guo

et al. (2023) parameterize the radiance field via a neural network and employ volumetric rendering techniques to reconstruct the 3D model from multi-view images. These representations interpret the specular reflection as the inherent appearance of the surface, enabling the photo-realistic rendering results. However, mistaking reflection for the base appearance of the object may lead to the sacrifice of geometry accuracy and limit the downstream task, *e.g.*, relighting. Besides implicit representation, recent 3D Gaussian splatting Kerbl et al. (2023); Huang et al. (2024); Jiang et al. (2023); Lu et al. (2024); Yu et al. (2024); Guédon & Lepetit (2024); Chen et al. (2023a); Yang et al. (2024); Lyu et al. (2024) involves iterative refinement of multiple Gaussians to reconstruct 3D objects from 2D images, allowing for the rendering of novel views in complex scenes through interpolation. It does not directly reconstruct the geometry but learns color and density in a volumetric point cloud. However, the inherently discrete representation of Gaussians also results in an inaccurate geometry, obstructing its wider applications. To improve the reconstructed geometry, surface-based methods Wang et al. (2021a); Li et al. (2023); Darmon et al. (2022); Oechsle et al. (2021); Vicini et al. (2022); Yariv et al. (2020); Wu et al. (2023); Yu et al. (2022); Sun et al. (2022); Liu et al. (2023a; 2024); Azinovic et al. (2022); Kirschstein et al. (2023) introduce a Signed Distance Field (SDF) to the volumetric representation, significantly enhancing the fidelity of geometry. Despite a more accurate surface representation, the misinterpretation of reflectance still exists due to the capacity of appearance network, affecting the learning of geometry.

## 2.2 Modeling Reflectance and Specularity

To well solve the problem of reflectance misinterpretation, several methods Liang et al. (2023); Wu et al. (2022); Guo et al. (2022); Boss et al. (2021); Zhang et al. (2021a;b; 2022); Jin et al. (2023); Tang et al. (2023a); Lv et al. (2023) employ the physical rendering equation to estimate the diffuse and specular components. Specifically, basis functions like spherical Gaussians Wang et al. (2009); Xu et al. (2013); Yariv et al. (2023); Zhang et al. (2021a) and spherical harmonics Fridovich-Keil et al. (2022); Basri & Jacobs (2003); Sloan et al. (2002); Yu et al. (2021a) are commonly used to better approximate rendering equation for a closed-form solution. However, the parameters of the basis functions are unknown and need to be learned by the neural network. These estimated parameters do not provide rendering-related information for the network during optimization. RefNeRF Verbin et al. (2022) instead introduces a reparametrization method to better distinguish reflectance from the appearance. Nevertheless, the reconstructed geometries are still undermined by the view-dependent optical phenomena. Following the reparametrized techniques, RefNeuS Ge et al. (2023) employs an anomaly detection technique for specularity to better reconstruct the geometry, but it produces inferior results for non-reflective objects. UniSDF Wang et al. (2023a) introduces a dual-branch structure to model both the reflective and non-reflective parts. It can reconstruct accurate shapes, but it fails to reconstruct high-frequency geometric details like thin structures. All these methods tackle only one-sided problems, either geometry or reflective appearance. Moreover, most methods designed for reflections always require instance-specific tuning. In contrast, our method improves geometry and appearance for both reflective and non-reflective surfaces, while avoiding instance-specific tuning.

## 3 Method

We first briefly review the neural implicit surface and the rendering equation to provide the basic background for this work (3.1). The reconstruction of geometry and appearance is a mutually reinforcing process. For the geometry, we design a fused-granularity neural surface to learn both shape and details, serving as a good base of appearance (3.2). For appearance, we incorporate the ASG encoding into a weight-modulated disentangled network to better interpret diffuse and specular color, reducing the ambiguity of geometry (3.3). Finally, we summarize our training objectives (3.4). The overview of our method is shown in Fig. 2.

### 3.1 Preliminaries

**Neural Implicit Surfaces.** NeRF Mildenhall et al. (2020) represents a 3D scene as volume density and color. Given a posed camera and a ray direction $d$, distance values $t_i$ are sampled along the corresponding ray $r = o + td$. The i-th sampled 3D position $x_i$ is then at a distance $t_i$ from the camera center. Spatial MLPs are then employed to map $x_i$ and $d$ to the volume density $\sigma_i$ and color

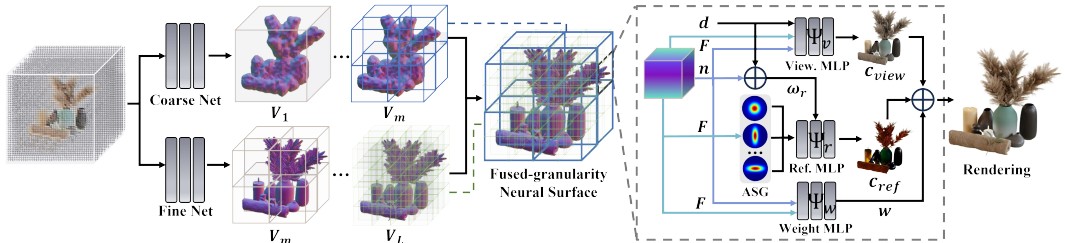

Figure 2: Pipeline of our method for 3D reconstruction. We utilize a fused-granularity neural surface structure where we make the most of coarse grids and fine grids for accurate surface reconstruction. We then employ a view-based radiance field and reflection-based radiance field to model diffuse part and specular part accordingly. By learning a 3D weight field, we blend the radiance fields to obtain high-fidelity renderings.

$c_i$ for prediction. The rendered color of a pixel is approximated as:

$$C = \sum_i w_i c_i, w_i = T_i \alpha_i, \tag{1}$$

where $\alpha_i = 1 - \exp(-\sigma_i \delta_i)$ is the opacity, $\delta_i = t_i - t_{i-1}$ is the distance between adjacent samples and $T_i = \Pi_{j=1}^{i-1} (1 - \alpha_j)$ is the accumulated transmittance. Despite that NeRF can reconstruct photo-realistic scenes, it is hard to extract surfaces using such density-based representations, leading to noisy and unrealistic results. To represent the scene geometry accurately, signed distance function (SDF) has been widely used as a surface representations. The surface $\mathcal{S}$ of an SDF can be represented by its zero-level set:

$$\mathcal{S} = \left\{ \mathbf{x} \in \mathbb{R}^3 \mid f(\mathbf{x}) = 0 \right\}, \tag{2}$$

where $f(\mathbf{x})$ is the SDF value. In the context of neural SDFs, NeuS Wang et al. (2021a) introduced SDF to the neural radiance fields with a logistic function to convert the SDF value to the opacity $\alpha_i$:

$$\alpha_i = \max \left( \frac{\Phi_s(f(\mathbf{x}_i)) - \Phi_s(f(\mathbf{x}_{i+1}))}{\Phi_s(f(\mathbf{x}_i))}, 0 \right), \tag{3}$$

where $\Phi_s$ is the sigmoid function. In this work, we adopt this SDF-based volume rendering formulation and optimize neural surfaces.

**Rendering Equations.** As introduced in Levoy & Hanrahan (1996), a light field can be defined as the radiance at a point in a given direction. A 5D function $L(\omega_o, \mathbf{x})$ can thus be used to represent the light field, where $\mathbf{x}$ is the position and $\omega_o$ is the outgoing radiance direction in spherical coordinates. This 5D light field is commonly modeled by employing the rendering equation:

$$\begin{aligned} L(\omega_o; \mathbf{x}) &= c_d + s \int_\Omega L_i(\omega_i; \mathbf{x}) \rho_s(\omega_i, \omega_o; \mathbf{x})(\mathbf{n} \cdot \omega_i) d\omega_i \\ &= c_d + s \int_\Omega f(\omega_i, \omega_o; \mathbf{x}, \mathbf{n}) d\omega_i = c_d + c_s, \end{aligned} \tag{4}$$

where $c_d$ represents the diffuse color and $s$ is the weight of the specular color $c_s$. $L_i$ is the incoming radiance from direction $\omega_i$, and $\rho_s$ represents the specular component of the spatially-varying bidirectional reflectance distribution function (BRDF). The function $f$ is defined to describe the outgoing radiance after the ray interaction. The final integral is solved over the hemisphere $\Omega$ defined by the normal vector $\mathbf{n}$ at point $\mathbf{x}$. Specifically, $L_i, \rho_s, \mathbf{n}$ are usually known functions or parameters that describe scene properties such as lighting, material, and shape. Following rendering equation, our method models diffuse and specularity using two radiance fields separately based on the viewing directions.

## 3.2 FUSED-GRANULARITY NEURAL SURFACES

Multi-resolution hash grid has proved its great scalability for generating fine-grained details, encouraging us to adopt it as the geometry representation. The hashgrids partition the space into blocks and

convey geometric information to the appearance networks. Despite fast convergence, it still suffers from a conflict that low-resolution grids produce over-smooth mesh but high-resolution grids induce overfitting. Through experiments, we have the following observations:

1. A coarser grid gets a larger partition with fewer blocks, which leads to easier convergence. A finer grid partitions the space with more blocks and requires longer training.

2. Using only coarse-grid leads to less-detailed results due to insufficient modeling ability. The limited ability of the hashgrid feature hinders the representation of detailed geometry.

3. Using only fine-grid leads to inaccurate results due to inaccurate learning of appearance network. At the early stage, before the appearance network disambiguates appearance, the fine-grid easily misinterprets specularity as redundant volumes, leading to noisy results.

4. Coarse-to-fine technique Wang et al. (2022); Li et al. (2023) improves overall details but may not preserve thin structures due to early insufficient partition of the coarse grids.

Based on these observations, we propose a fused-granularity structure to consider the fitting nature of hashgrids for detailed reconstruction. The fused-granularity neural surfaces initialize and train a set of coarse-granularity grids and a set of fine-granularity grids together and progressively. Coarse-grids converge faster at the early training stage, we then ensure that fine-grids remain in close proximity to the coarse-grids by restricting the normals using curvature loss. Fine-grids can fit the details by smaller partitions as training continues.

Specifically, we first define $\{V_1, \ldots, V_m\}$ to be the coarse-granularity set and $\{V_m, \ldots, V_L\}$ to be the fine-granularity set of multi-resolution hash grids. Given an input position $\mathbf{x}_i$, we employ coarse-to-fine methods to map it to each grid resolution $V_l$ to get $\mathbf{x}_{i,l}$ in both granularity sets separately. Then the feature vector $\gamma_l$ given resolution $V_l$ is obtained via trilinear interpolation of hash entries. The encoding features are then concatenated together as:

$$\gamma^c(\mathbf{x}_i) = (\gamma_1(\mathbf{x}_{i,1}), \ldots, \gamma_m(\mathbf{x}_{i,m})),$$
$$\gamma^f(\mathbf{x}_i) = (\gamma_m(\mathbf{x}_{i,m}), \ldots, \gamma_L(\mathbf{x}_{i,L})),$$

(5)

where the resolution level $m$ and $L$ are set empirically. The encoded features $\gamma^c$ and $\gamma^f$ serve as the inputs to corresponding branch-MLPs that predict the SDF values and geometric features. The SDF values and the geometric features of two branches are then fused into a single set of values that are passed to the appearance network:

$$SDF = SDF^c + SDF^f,$$
$$F = F^c + F^f.$$

(6)

The fused-granularity structure can effectively avoid discarding thin structures in the early stage, as the fine-granularity grids do not continue from coarse-granularity grids but start from a higher-resolution initialization.

### 3.3 BLENDED RADIANCE FIELDS WITH ASG ENCODING

Estimating color directly using a radiance field usually results in inaccurate geometry for reflective surfaces due to the misinterpretation of the reflectance. Consequently, the MLP is burdened with learning the complex physical meanings of the rendering equation, posing a considerable challenge. Several methods instead predict the parameters of basis functions like spherical Gaussians and spherical harmonics to estimate the color. Nevertheless, these parameters do not convey much rendering-related information to the network and thus cannot represent high-frequency appearance details. In order to disambiguate geometry, color, and reflections, the appearance network should have the capacity to represent both diffuse and specular parts. Following Eq. 4, we design a blended radiance field structure to model the diffuse and specularity separately. A reparametrized technique Verbin et al. (2022); Ge et al. (2023); Yariv et al. (2023) is typically adopted to model the reflection viewing direction:

$$\omega_r = 2(-\mathbf{d} \cdot \mathbf{n}) \cdot \mathbf{n} + \mathbf{d},$$

(7)

where the normal can be derived as $\mathbf{n} = \nabla d(x)/\|\nabla d(x)\|$ from the signed distance $d(x)$. Unfortunately, it cannot balance the general non-reflective surfaces due to the misalignment of physically accurate normals. Therefore, we use it to only model the specular components.

Compared with fixed-basis encodings like SHs and SGs, anisotropic spherical Gaussian (ASG) Xu et al. (2013); Han & Xiang (2023) can attain more comprehensive encoding, enabling the representation of full-frequency signals. Due to its ability to represent high-frequency details, we employ the ASG to encode Eq. 4 in the feature space:

$$ASG(\omega_o \mid [x,y,z], [\lambda, \mu], \xi) = \xi \cdot \mathbf{S}(\omega_o; z) \cdot e^{-\lambda(\omega_o \cdot x)^2 - \mu(\omega_o \cdot y)^2}, \tag{8}$$

where $[x, y, z]$ (lobe, tangent and bi-tangent) are predefined orthonormal axes in ASG. $\lambda \in \mathbb{R}^1$ and $\mu \in \mathbb{R}^1$ represents the sharpness parameters controlling the shape of ASG. $\xi \in \mathbb{R}^2$ represents the lobe amplitude and S is the smooth term defined as $\mathbf{S}(\omega_o; z) = \max(\omega_o \cdot z, 0)$. We first learn the anisotropic information as a latent feature and pass the feature to the reflection MLP $\Psi_r$ to take advantage of the encoded rendering equation, where $\Psi_r$ is then used to predict the integrated color from the resultant encoding instead of approximating a complex function. We derive the ASG-encoded feature as follows:

$$\begin{aligned} \lambda, \mu, \xi &= f_{par}(F, \mathbf{n}), \\ F_{asg}^i &= ASG(\omega_r \mid [x,y,z], [\lambda_i, \mu_i], \xi_i), \\ F_{asg} &= [F_{asg}^1, F_{asg}^2, \cdots, F_{asg}^N], \end{aligned} \tag{9}$$

where the parameters $\lambda$, $\mu$, and $\xi$ in our model are learned by a compact network $f_{par}$.

Overall, given the 3D position $\mathbf{x}$ and view-direction $d$, our blended radiance fields can be summarized as:

$$\begin{aligned} c_{view} &= \Psi_v(\mathbf{x}, \mathbf{d}, \mathbf{n}, F), \\ c_{ref} &= \Psi_r(F_{asg}, \omega_r), \end{aligned} \tag{10}$$

where $n$ is the normal at position $x$, $F$ is the geometric features from the previous SDF MLP. $\omega_r$ here is the reparametrized reflected viewing direction and $\Psi_v$, $\Psi_r$ are the MLPs for view-based radiance field and reflection-based radiance field. By employing the ASG encoding in this branch, AniSDF can model scenes with complex appearances. Furthermore, it is noteworthy that we learn the geometry based on pixel-level supervision. Once the representing ability of the appearance network is enhanced on the pixel level, the geometry network is more likely to capture high-frequency details on the geometry level. Inspired by UniSDF Wang et al. (2023a), the blended radiance fields are composed using a learned 3D weight field:

$$w = \Phi_s(\Psi_w(\mathbf{x}, \mathbf{n}, F)), \tag{11}$$

where $\Phi_s$ is the sigmoid function. The two radiance fields are then composed at the pixel level:

$$C = w * c_{view} + (1 - w) * c_{ref}. \tag{12}$$

## 3.4 Loss Functions

Our model utilizes the RGB loss between the rendered color and the ground-truth color during the training process:

$$\mathcal{L}_{rgb} = ||C - C_{gt}||^2. \tag{13}$$

Following prior surface reconstruction works, we adopt the Eikonal loss in order to better approximate a valid SDF:

$$\mathcal{L}_{eik} = \mathbb{E}_\mathbf{x} \left[ (\|\nabla f(\mathbf{x})\| - 1)^2 \right]. \tag{14}$$

To encourage the model to learn smooth surfaces, we also adapt the curvature loss proposed by PermutoSDF Rosu & Behnke (2023) to our fused-granularity neural surfaces:

$$\mathcal{L}_{curv} = \sum_x (\mathbf{n} \cdot \mathbf{n}_\epsilon - 1)^2, \tag{15}$$

where $n$ is the normal at each position and $n_\epsilon$ is obtained by slight perturbation of the sample $\mathbf{x}$. We also employ the orientation loss Barron et al. (2021) to penalize the "back-facing" normals:

$$\mathcal{L}_o = \sum_i w_i \max(0, \mathbf{n} \cdot \mathbf{d})^2. \tag{16}$$

For finer geometric details that align with physically correct representation, we regularize the transparency $\alpha$ to be either 0 or 1:

$$\mathcal{L}_\alpha = BCE(\alpha, \alpha), \tag{17}$$

where $BCE$ refers to the binary cross entropy loss.

Overall, the full loss function in our model is defined to be:

$$\mathcal{L} = \mathcal{L}_{rgb} + \lambda_1 \mathcal{L}_{eik} + \lambda_2 \mathcal{L}_{curv} + \lambda_3 \mathcal{L}_o + \lambda_4 \mathcal{L}_\alpha. \tag{18}$$

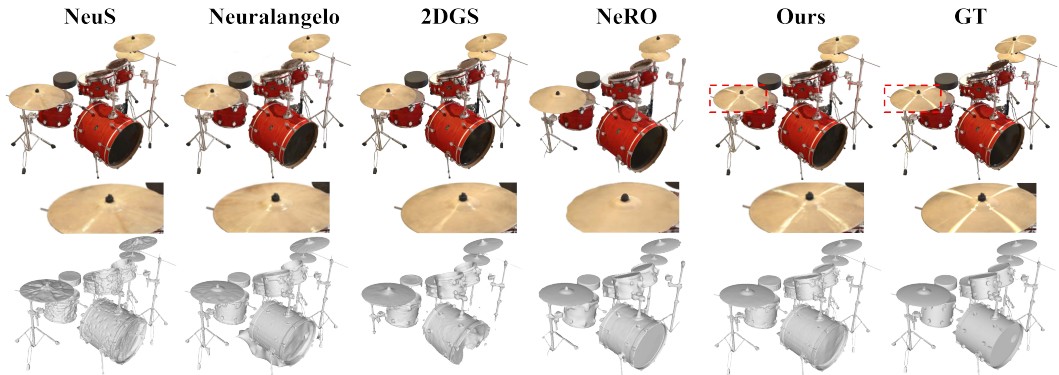

Figure 3: Comparison on NeRF synthetic dataset with previous surface reconstruction methods. Our model yields the most accurate geometry reconstruction and highest-quality rendering at the same time. Our model can handle the semi-transparent structure and produce accurate renderings for the specular parts.

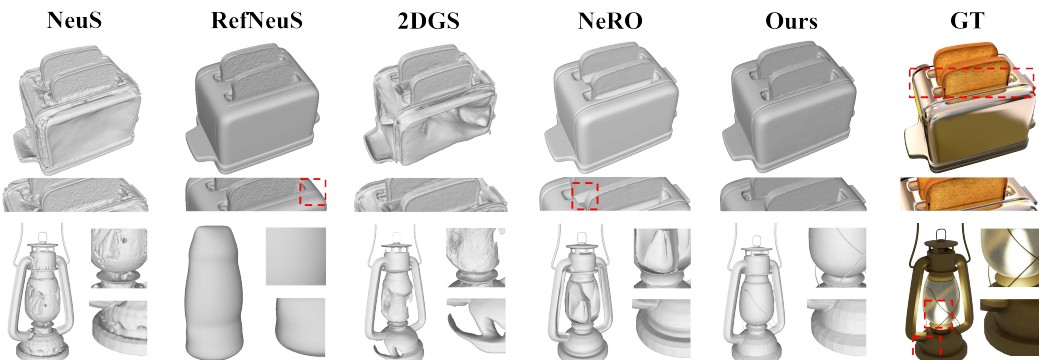

Figure 4: Comparison on Shiny Blender dataset with previous surface reconstruction methods. Our model achieves the most accurate surface reconstruction for reflective objects. In addition, our method can reconstruct luminous objects while all the other methods fail to reconstruct surfaces.

## 4 EXPERIMENTS

### 4.1 EXPERIMENT SETUPS

In our experiment, we use NeRF Synthetic dataset Mildenhall et al. (2020), DTU dataset Wang et al. (2021a), Shiny Blender dataset Verbin et al. (2022), Shelly dataset Wang et al. (2023d) for training and evaluation. We also construct a luminous dataset to demonstrate the ability of our method. Our model is trained using a single Tesla V100 for around 2-3 hours and the hyperparameters for the loss function in our method are set to be: $\lambda_1 = 0.1, \lambda_2 = 0.001, \lambda_3 = 0.001, \lambda_4 = 0.01$. Our coarse-grid is from level 4 to 10 $(m)$, and fine-grid is from 10 $(m)$ to 16 $(L)$, both with 2 as feature dimension. We learn these two parallel hashgrids without increasing the gridsize that leads to high memory consumption. Both the geometry network MLP and View.MLP have 2 hidden layers with 64 neurons. The Ref.MLP has 2 hidden layers with 128 neurons and the Weight.MLP has 1 hidden layers with 64 neurons. We use marching cubes as the mesh extraction tools.

### 4.2 COMPARISONS

**NeRF Synthetic Dataset.** We compare the reconstruction results on NeRF Synthetic Dataset Mildenhall et al. (2020) with previous surface reconstruction methods as shown in Fig. 3. The corresponding qualitative evaluation results are displayed in Table. 1. It can be seen that our method achieves high-quality rendering with the most accurate geometry. With the ASG encoding used in the blended radiance field, our method can produce reflective details, *e.g.*, reflection on the

| | | | Chair | Drums | Ficus | Hotdog | Lego | Materials | Mic | Ship | Avg |
|---|---|---|---|---|---|---|---|---|---|---|---|
| PSNR↑ | Volumetric | NeRF | 34.17 | 25.08 | 30.39 | 36.82 | 33.31 | 30.03 | 34.78 | 29.30 | 31.74 |
| | | InstantNGP | 35.00 | 26.02 | 33.51 | 37.40 | 36.39 | 29.78 | 36.22 | 31.10 | 33.18 |
| | | Mip-NeRF | 35.14 | 25.48 | 33.29 | 37.48 | 35.70 | 30.71 | 36.51 | 30.41 | 33.09 |
| | | Zip-NeRF | 34.84 | 25.84 | 33.90 | 37.14 | 34.84 | 31.66 | 35.15 | 31.38 | 33.10 |
| | | 3DGS | 35.36 | 26.15 | 34.87 | 37.72 | 35.78 | 30.00 | 35.36 | 30.80 | 33.32 |
| | Surface | NeuS | 31.22 | 24.85 | 27.38 | 36.04 | 34.06 | 29.59 | 31.56 | 26.94 | 30.20 |
| | | NeRO | 28.74 | 24.88 | 28.38 | 32.13 | 25.66 | 24.85 | 28.64 | 26.55 | 27.48 |
| | | BakedSDF | 31.65 | 20.71 | 26.33 | 36.38 | 32.69 | 30.48 | 31.52 | 27.55 | 29.66 |
| | | NeRF2Mesh | 34.25 | 25.04 | 30.08 | 35.70 | 34.90 | 26.26 | 32.63 | 29.47 | 30.88 |
| | | 2DGS | 35.05 | 26.05 | 35.57 | 37.36 | 35.10 | 29.74 | 35.09 | 30.60 | 33.07 |
| | | Ours | 35.31 | 26.23 | 33.15 | 37.99 | 35.69 | 31.87 | 35.44 | 31.69 | 33.42 |
| Chamfer Distance↓ | Surface | NeuS | 3.95 | 6.68 | 2.84 | 8.36 | 6.62 | 4.10 | 2.99 | 9.54 | 5.64 |
| | | NeRF2Mesh | 4.60 | 6.02 | 2.44 | 5.19 | 5.85 | 4.51 | 3.47 | 8.39 | 5.06 |
| | | NeRO | 3.66 | 8.25 | 10.52 | 4.79 | 8.93 | 5.68 | 3.65 | 21.05 | 8.32 |
| | | BakedSDF | 4.05 | 7.41 | 3.23 | 6.72 | 5.69 | 5.39 | 3.17 | 8.98 | 5.58 |
| | | Neuralangelo | 14.50 | 16.99 | 5.72 | 14.27 | 6.90 | 3.27 | 8.78 | 16.02 | 10.81 |
| | | 2DGS | 5.25 | 10.33 | 4.41 | 9.55 | 6.74 | 9.09 | 11.06 | 9.55 | 8.25 |
| | | Ours | 4.39 | 5.24 | 2.75 | 7.81 | 5.16 | 3.03 | 5.34 | 5.41 | 4.89 |

Table 1: Quantitative comparison on NeRF Synthetic dataset. We compare our model with previous volumetric rendering methods and surface-reconstruction methods, with each cell colored to indicate the best and second. Our method achieves the hightest quality in both novel view synthesis and surface reconstruction with the highest PSNR ↑ and lowest Chamfer Distance ↓ (with $10^{-3}$ as the unit).

| Methods | Helmet | | Toaster | | Coffee | | Car | | Mean | |
|---|---|---|---|---|---|---|---|---|---|---|
| | PSNR↑ | MAE↓ | PSNR↑ | MAE↓ | PSNR↑ | MAE↓ | PSNR↑ | MAE↓ | PSNR↑ | MAE↓ |
| NeuS | 27.78 | 1.12 | 23.51 | 2.87 | 28.82 | 1.99 | 26.34 | 1.10 | 26.61 | 1.77 |
| RefNeRF | 29.68 | 29.48 | 25.70 | 42.87 | 34.21 | 12.24 | 30.82 | 14.93 | 30.10 | 24.88 |
| RefNeuS | 32.85 | 0.38 | 26.97 | 1.47 | 31.05 | 0.99 | 29.92 | 0.80 | 30.20 | 0.91 |
| Ours | 34.44 | 0.41 | 26.98 | 1.15 | 33.24 | 1.14 | 29.56 | 0.70 | 31.05 | 0.85 |

Table 2: Quantitative comparison on Shiny Blender dataset with each cell colored to indicate the best and second. We compare our approach with previous reflective surface reconstruction methods. Our model achieves the best results in both novel view synthesis and surface reconstruction with the highest PSNR ↑ and lowest surface normal mean angular error MAE ↓.

gong, while other methods fail to synthesize the complex specularity. Thanks to the proposed fused-granularity surfaces, our method also outperforms others on the high-frequency geometric details, *e.g.*, the net of sails on the ship.

**Shiny Blender Dataset.** To further demonstrate the positive effect of the ASG encoding on the geometry, we compare the geometry of our method with previous reflective surface reconstruction methods on the Shiny Blender Dataset Verbin et al. (2022), as shown in Fig. 4. The corresponding quantitative results are also provided in Table. 2. NeuS Wang et al. (2021a) and 2DGS Huang et al. (2024) suffer from the ambiguity of reflective surfaces and synthesize a concave surface of the toaster. RefNeuS Ge et al. (2023) and NeRO Liu et al. (2023b) release the problem of reflective ambiguity, but their geometries are smooth and lack details. Besides, they also have artifacts, *e.g.*, a missing handle for RefNeuS, and a hole of bread reflection for NeRO. Due to the better modeling of diffuse and specular appearance, our architecture can better represent the concavity and convexity of a reflective object, and further solve the ambiguity of surfaces. We also demonstrate high-quality results in both the rendering and geometry of reflective objects in quantitative results in Table. 2.

**DTU Dataset.** We also compare our method with previous SDF-based methods on the DTU dataset that involves the ground truth of the point cloud, more suitable for geometry comparison. The qualitative comparison results are shown in Table. 3. Our method achieves the best results among all methods.

**Complex Objects.** Moreover, we provide more complex cases on the Shelly Datasets Wang et al. (2023d) to further demonstrate the ability of our model to reconstruct fuzzy objects. As shown in

| Scan ID | 24 | 37 | 40 | 55 | 63 | 65 | 69 | 83 | 97 | 105 | 106 | 110 | 114 | 118 | 122 | Mean |
|---|---|---|---|---|---|---|---|---|---|---|---|---|---|---|---|---|
| COLMAP | 0.81 | 2.05 | 0.73 | 1.22 | 1.79 | 1.58 | 1.02 | 3.05 | 1.40 | 2.05 | 1.00 | 1.32 | 0.49 | 0.78 | 1.17 | 1.36 |
| NeRF | 1.90 | 1.60 | 1.85 | 0.58 | 2.28 | 1.27 | 1.47 | 1.67 | 2.05 | 1.07 | 0.88 | 2.53 | 1.06 | 1.15 | 0.96 | 1.49 |
| NeuS | 1.00 | 1.37 | 0.93 | 0.43 | 1.10 | 0.65 | 0.57 | 1.48 | 1.09 | 0.83 | 0.52 | 1.20 | 0.35 | 0.49 | 0.54 | 0.84 |
| VolSDF | 1.14 | 1.26 | 0.81 | 0.49 | 1.25 | 0.70 | 0.72 | 1.29 | 1.18 | 0.70 | 0.66 | 1.08 | 0.42 | 0.61 | 0.55 | 0.86 |
| Neuralangelo | 0.49 | 1.05 | 0.95 | 0.38 | 1.22 | 1.10 | 2.16 | 1.68 | 1.78 | 0.93 | 0.44 | 1.46 | 0.41 | 1.13 | 0.97 | 1.07 |
| NeuralWarp | 0.49 | 0.71 | 0.38 | 0.38 | 0.79 | 0.81 | 0.82 | 1.20 | 1.06 | 0.68 | 0.66 | 0.74 | 0.41 | 0.63 | 0.51 | 0.68 |
| Gaussian Surfels | 0.66 | 0.93 | 0.54 | 0.41 | 1.06 | 1.14 | 0.85 | 1.29 | 1.53 | 0.79 | 0.82 | 1.58 | 0.45 | 0.66 | 0.53 | 0.88 |
| 2DGS | 0.48 | 0.91 | 0.39 | 0.39 | 1.01 | 0.83 | 0.81 | 1.36 | 1.27 | 0.76 | 0.70 | 1.40 | 0.40 | 0.76 | 0.52 | 0.80 |
| Ours | 0.52 | 0.82 | 0.65 | 0.43 | 0.76 | 0.64 | 0.71 | 0.97 | 0.86 | 0.64 | 0.52 | 0.67 | 0.42 | 0.67 | 0.50 | 0.65 |

Table 3: Quantitative comparison on the DTU dataset with each cell colored to indicate the best and second . We compare our method with previous surface-reconstruction methods. Our method achieves the highest quality of surface reconstruction with the lowest Chamfer Distance↓.

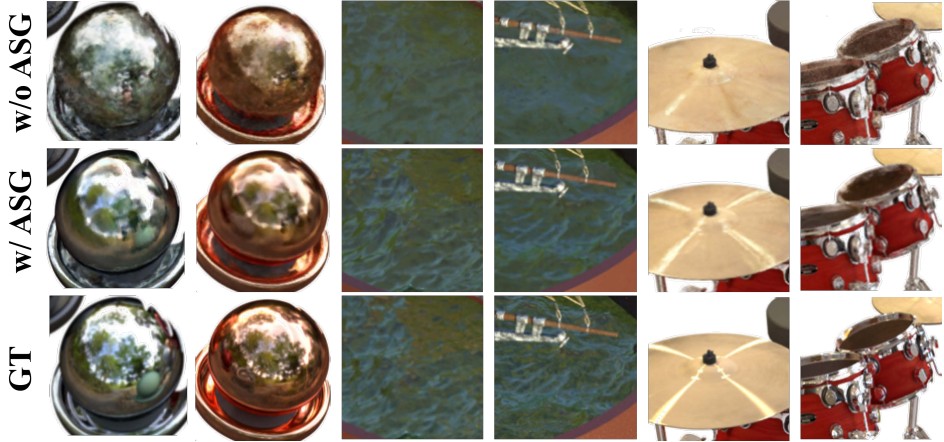

Figure 5: Ablation results on ASG encoding. We demonstrate the ability to synthesize specular details with the use of ASG encoding.

Fig. 19, 2DGS produces blurry results, while our method successfully reconstructs the details of hair and fur. Additionally, we build a luminous dataset and compare various methods on it. We display an extremely hard case with thin lines and luminous glass in Fig. 4. RefNeuS Ge et al. (2023) generates a coarse and smooth mesh without any details. Despite more details, NeuS Wang et al. (2021a) and 2DGS Huang et al. (2024) produce a broken shell. NeRO Liu et al. (2023b) learns a relatively complete shape but performs badly in the luminous part and fails to reconstruct the thin lines. In contrast, our model can disambiguate the geometry from the luminous appearance and generate accurate geometry of both structures.

## 4.3 ABLATION STUDIES

In this section, we study the effect of individual components proposed in our work, *i.e.*, Anisotropic Spherical Gaussian (ASG) encoding, and fused-granularity neural surfaces.

**ASG Encoding.** We compare ASG encoding with common positional encoding based on the same geometry learning pipeline. As demonstrated in Fig. 5, the results with ASG encoding show better appearances with clearer reflections compared to the positional encoding. This is because ASG encoding can represent anisotropic scenes more comprehensively making better use of the rendering equation than other fixed basis-functions encoding. We also show quantitative experiments conducted on the NeRF synthetic dataset in Table. 4. It also demonstrates the superiority of ASG encoding.

**Fused-Granularity Neural Surface.** To prove the effectiveness of fused-granularity surface, we set the same appearance learning structure and compare our architecture with previous single-branch coarse-to-fine architecture. As shown in Fig. 6, results without fused granularity miss details in high-frequency parts. Due to the initialization from a coarse grid, it filters the thin structure like

|  | Rendering (PSNR↑) | Geometry (CD↓) |
|---|---|---|
| w/o ASG, w/o Fused | 30.25 | 5.64 |
| w/o ASG, w/ Fused | 32.38 | 5.16 |
| w/ ASG, w/o Fused | 33.19 | 5.37 |
| **w/ ASG, w/ Fused (Ours)** | 33.42 | 4.89 |

Table 4: Ablation quantitative results on NeRF Synthetic dataset. We demonstrate the effectiveness of the ASG encoding and fused-granularity surfaces.

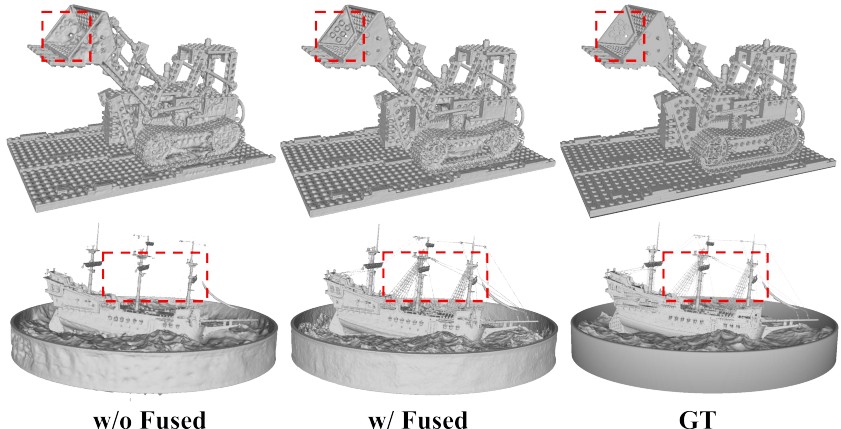

| w/o Fused | w/ Fused | GT |

Figure 6: Ablation on fused-granularity neural surfaces. We show the effectiveness of the fused-granularity surfaces for geometric details reconstruction.

the net of sails in the early stage and cannot recover filtered parts in the following fine stage. In contrast, because of an additional fine grid initialization, results with fused granularity can keep thin structures in the early stage and reconstruct them during the fine stage. The quantitative experiments in Table. 4 show the consistent results with the qualitative results. Besides, with both the ASG encoding and fused-granularity surfaces, we can obtain the best results.

## 5 CONCLUSION

In this work, we present AniSDF, a unified SDF-based approach that optimizes fused-granularity neural surfaces with anisotropic encoding for high-fidelity 3D reconstruction. Our method is based on two key components: 1) Fused-granularity neural surfaces that make the most of both coarse-granularity hash grids and fine-granularity hash grids. 2) Blended radiance fields that blend the view-based radiance field and reflection-based radiance field with anisotropic spherical Gaussian encoding. The first component enables the representation of high-frequency geometric details and balances the overall structures and high-frequency geometric details. The second component takes advantage of the rendering equation and allows our model to synthesize photorealistic renderings, successfully disambiguating the reflective appearance. Extensive experiments showcase that our method achieves high-quality results in both geometry reconstruction and novel-view synthesis.

## 6 LIMITATIONS

Despite high-quality results, AniSDF still has several limitations. (1) AniSDF cannot achieve real-time rendering. It could be a possible solution that we adapt the SDF-baking method from BakedSDF Yariv et al. (2023) for ASG encoding to improve the efficiency in future work. (2) Another limitation is that AniSDF fails in cases with complex indirect illumination due to the lack of a materials estimation network.

## 7 ACKNOWLEDGEMENT

This work was supported in part by NSFC (62201342), and Shanghai Municipal Science and Technology Major Project (2021SHZDZX0102). Authors would like to appreciate the Student Innovation Center of SJTU for providing GPUs.

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

# A   APPENDIX

## A.1   FUSED-GRANULARITY NEURAL SURFACES ABLATION

To further demonstrate the effectiveness of the fused-granularity structure, we conduct additional experiments on several related components including **1.** maximum hashgrid resolution, **2.** different granularity structure, **3.** experimental results clarification.

**Maximum hashgrid resolution.** As shown in Fig. 7, we vary the maximum hashgrid resolution from 1024 to 4096. It can be seen that by using 1024 as the maximum resolution, the model can not reconstruct some geometric details like the net of the sails. On the other hand, using 4096 as the maximum resolution can achieve slightly better results but requires larger memory consumption (32GB). Therefore, we use 2048 as the default resolution in our model (24GB).

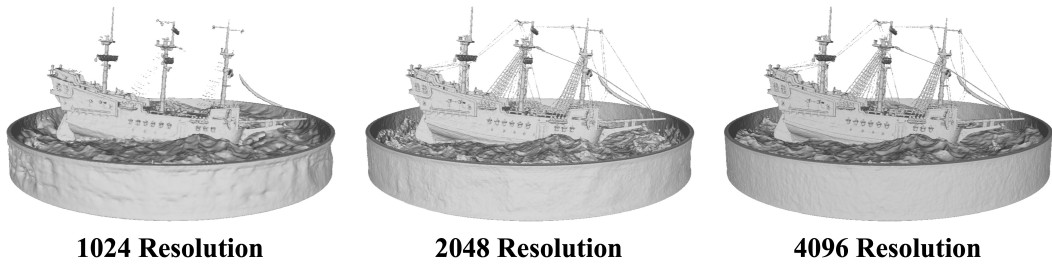

**1024 Resolution**              **2048 Resolution**              **4096 Resolution**

Figure 7: Ablation of maximum hashgrid resolution. We showcase the *ship* scene reconstructed using different maximum hashgrid resolution.

**Different granularity structure.** As shown in Fig. 8, results with two fine branches with the same resolution setting produce a noisy surface and require larger memory consumption (30GB), while those with two coarse branches fail to reconstruct geometric details like the net of the sails. In contrast, the fused-granularity structure make the most of both branches and reconstruct the best results with moderate memory consumption (24GB).

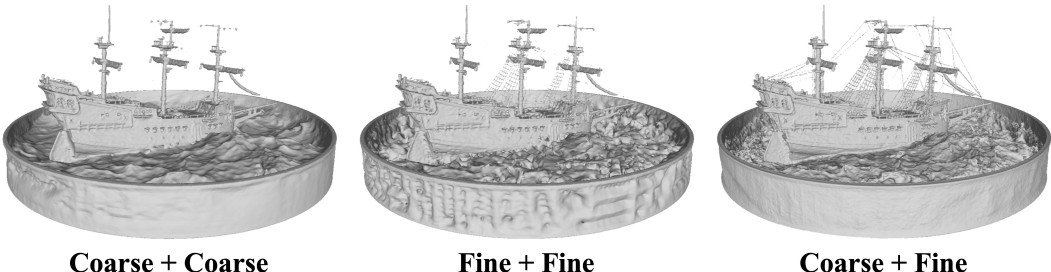

**Coarse + Coarse**              **Fine + Fine**              **Coarse + Fine**

Figure 8: Ablation of granularity structure. We showcase the *ship* scene reconstructed using different granularity structure.

**Experimental results clarification.** We showcase the ablation study of level $m$ in Fig. 9. It can be seen that setting $m = 9$ and $m = 11$ both generate inferior results. We also present the comparison of the reconstructed results of *mic* in Fig. 10. Since most methods tend to learn a solid geometry for those objects with hollow parts, e.g. Mic., Chamfer Distance is not always accurate to reflect the geometry quality. Due to its points sampling process on the mesh, it tends to produce better quantitative results for a smooth surface for these objects. Our reconstructed geometry displays better visual effect than NeRO but gets a worse quantitative result. We also conduct an experiment by adjusting the hyper-parameters to produce a smoother surface and achieve the best metric.

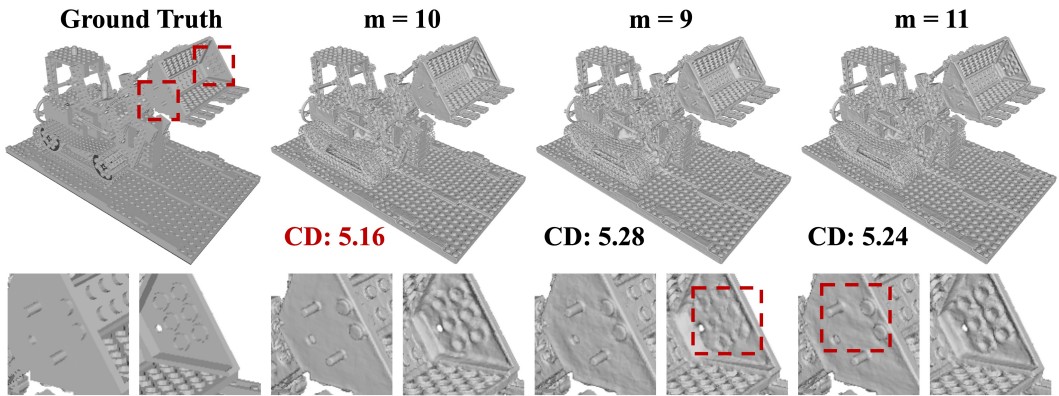

Figure 9: Ablation of level $m$. We showcase the *lego* scene reconstructed using different level $m$.

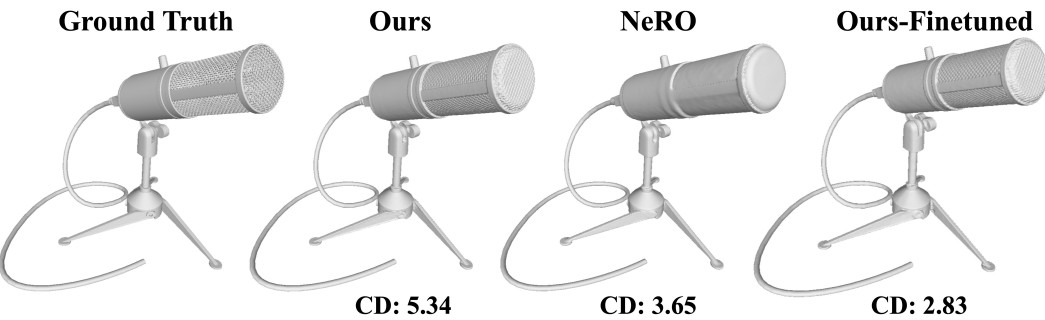

Figure 10: Comparison on reconstruction results of *mic*.

## A.2 BLENDED RADIANCE FIELDS VISUALIZATION.

AniSDF utilizes blended radiance fields to reconstruct high-fidelity renderings. As shown in Fig. 11, our method can well separate reflectance from base color by utilizing the blended radiance fields structure. AniSDF also has the ability to reconstruct reflective surfaces for real-world cases as shown in Fig. 12, .

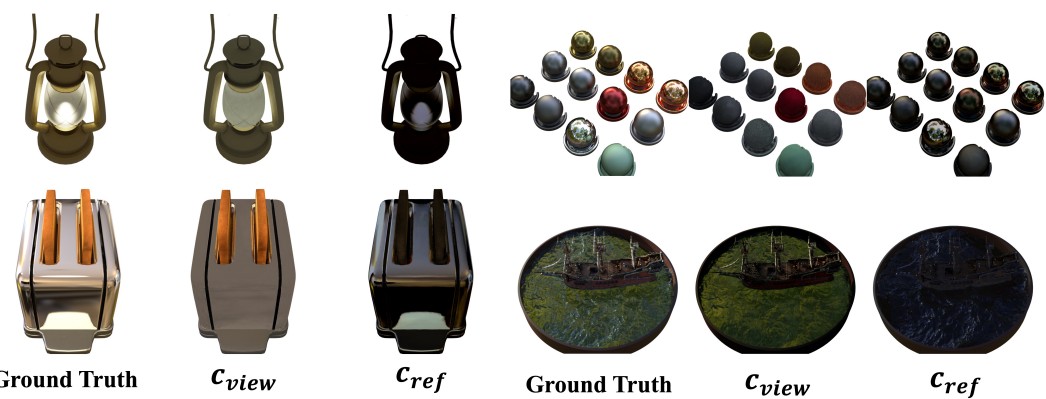

Figure 11: Visualization of $c_{view}$ and $c_{ref}$. We showcase the reconstructed results of $c_{view}$ and $c_{ref}$ by our blended radiance fields.

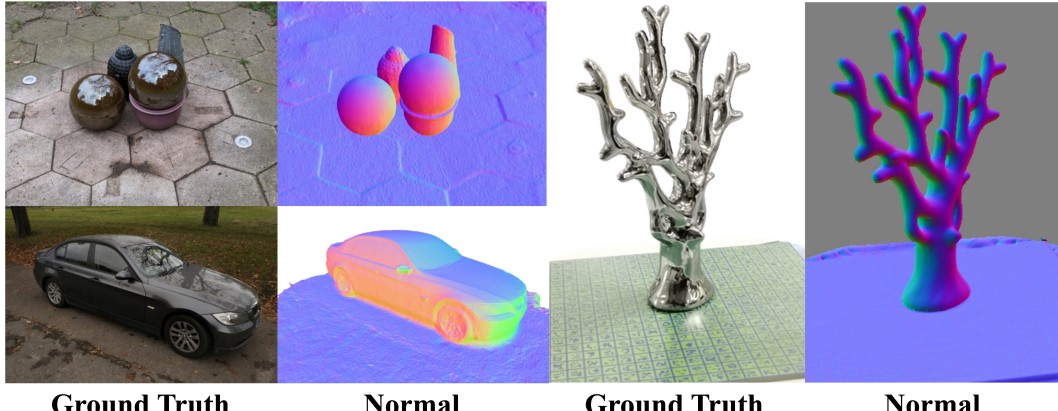

**Ground Truth**       **Normal**       **Ground Truth**       **Normal**

Figure 12: Real-world reflective surface reconstruction results. We showcase the reconstruction results of *sedan*,*gardensphere* in Shiny Blender dataset and *coral* in NeRO Glossy-Real dataset.

## A.3 UNBOUNDED SCENES

AniSDF can also reconstruct real unbounded scene with great details. We use MipNeRF360 Barron et al. (2021) for demonstration. The reconstructed results are shown in Fig. 13. Our method can reconstruct accurate geometry including the thin structures and the fuzzy background with high-fidelity rendering. We also present the foreground rendering result with the depth map and normal map of the *bicycle* and *bonsai* scene in Fig. 14. The qualitative comparison of rendering quality are shown in Table. 5. By employing the fused-granularity neural surfaces along with the anisotropic encoding, we can synthesize high-quality rendering for real-life complex scenes.

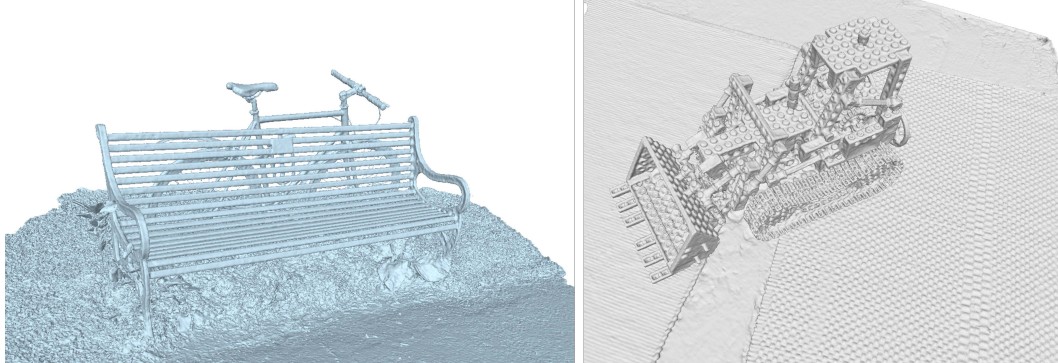

Figure 13: Reconstructed mesh results of MipNeRF360 dataset. We showcase the *bicycle* and *kitchen* scene reconstructed using our method.

| Method | Outdoor | | | | | Avg. | Indoor | | | | Avg. |
|---|---|---|---|---|---|---|---|---|---|---|---|
| | bicycle | flowers | garden | stump | treehill | | room | counter | kitchen | bonsai | |
| InstantNGP | 22.79 | 19.19 | 25.26 | 24.80 | 22.46 | 22.90 | 30.31 | 26.21 | 29.00 | 31.08 | 29.15 |
| Mip-NeRF | 24.40 | 21.64 | 26.94 | 26.36 | 22.81 | 24.47 | **31.40** | 29.44 | 32.02 | 33.11 | **31.72** |
| 3DGS | 25.24 | 21.52 | 27.41 | 26.55 | 22.49 | 24.64 | 30.63 | 28.70 | 30.32 | 31.98 | 30.41 |
| BakedSDF | 23.05 | 20.55 | 26.44 | 24.39 | 22.55 | 23.40 | 30.68 | 27.99 | 30.91 | 31.26 | 30.21 |
| UniSDF | 24.67 | 21.83 | 27.46 | 26.39 | **23.51** | 24.77 | 31.25 | 29.26 | **31.73** | 32.86 | 31.28 |
| 2DGS | 24.87 | 21.15 | 26.95 | 26.47 | 22.27 | 24.34 | 31.06 | 28.55 | 30.50 | 31.52 | 30.40 |
| Ours | **25.36** | **22.32** | **27.65** | **26.63** | 23.02 | **24.99** | 31.30 | **30.23** | 31.69 | **33.25** | 31.62 |

Table 5: Rendering comparison on MipNeRF360 dataset. We compare our method with previous methods and present the PSNR↑ results.

| Foreground | Depth | Normal |
|---|---|---|

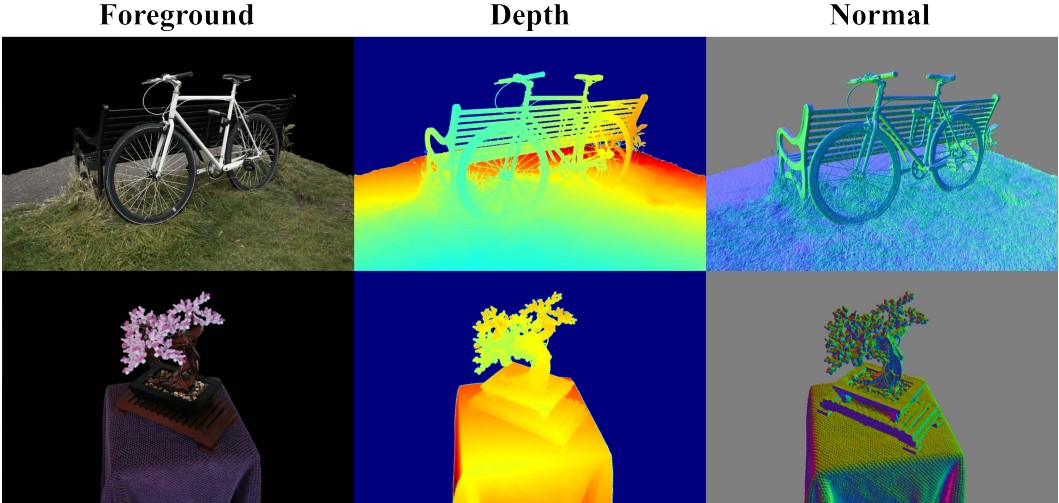

Figure 14: Real unbounded scene reconstruction results of MipNeRF360 dataset. We showcase the *bicycle* and *bonsai* scene reconstructed using our method.

## A.4 Additional Experimental Results

AniSDF reconstructs high-fidelity geometry results while boosting the rendering quality of SDF-based methods by a great scale. We present an additional comparison on the *Ship* scene in NeRF synthetic dataset in Fig. 15. Our model yields the most accurate geometry reconstruction and highest-quality rendering at the same time. Our model can handle the net-like structure and produce accurate renderings for the specular parts.

| NeuS | Neuralangelo | 2DGS | NeRO | Ours | GT |
|---|---|---|---|---|---|

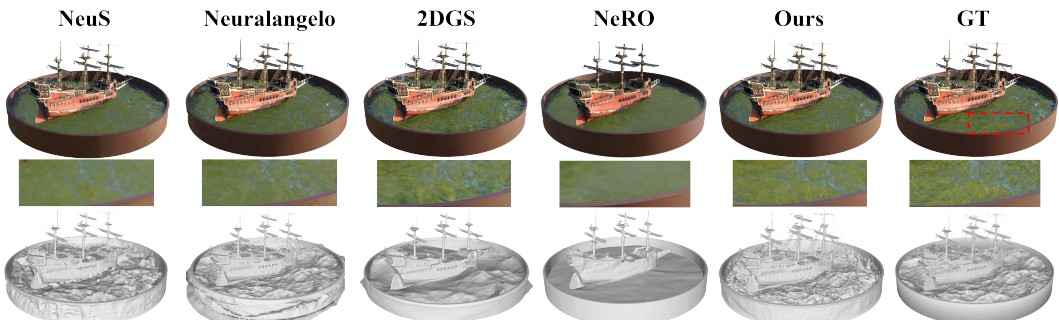

Figure 15: Comparison on NeRF synthetic dataset with previous surface reconstruction methods.

We also present additional results to demonstrate the highly detailed mesh reconstructed using our method in Fig. 16. To further demonstrate the reconstruction of thin structures, we present the rendering result along with the depth map and normal map in Fig. 17. It can be seen that we can reconstruct thin structures and synthesize high-frequency renderings like the reflection.

We showcase the reconstruction of fuzzy object in Fig. 18 and Fig. 19. It is noteworthy that reconstructing hair is a long-standing challenging problem for surface reconstruction methods. Nevertheless, our model yields a more accurate representation than the other methods. Our method can also synthesize high-fidelity renderings for hair and fur that surpass all the surface-based methods.

For DTU dataset, as shown in Fig. 20 and Fig. 21, our method can reconstruct accurate surfaces for objects with rich details while other methods introduce noise or oversmooth results. In addition, our method can reconstruct accurate reflective surfaces under complex lighting.

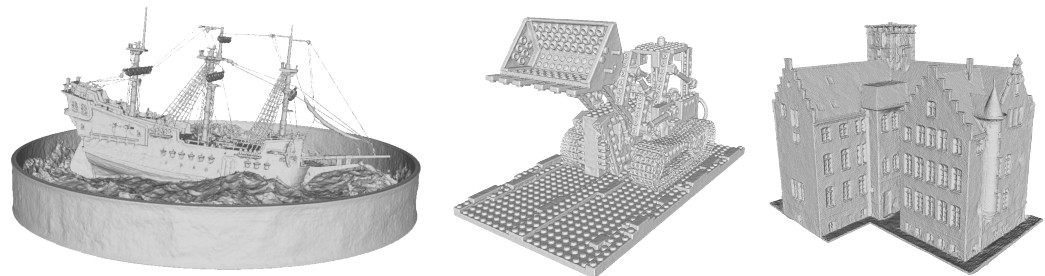

Figure 16: Detailed presentation of the reconstructed mesh by our method. We demonstrate that we can reconstruct accurate geometry with fine details.

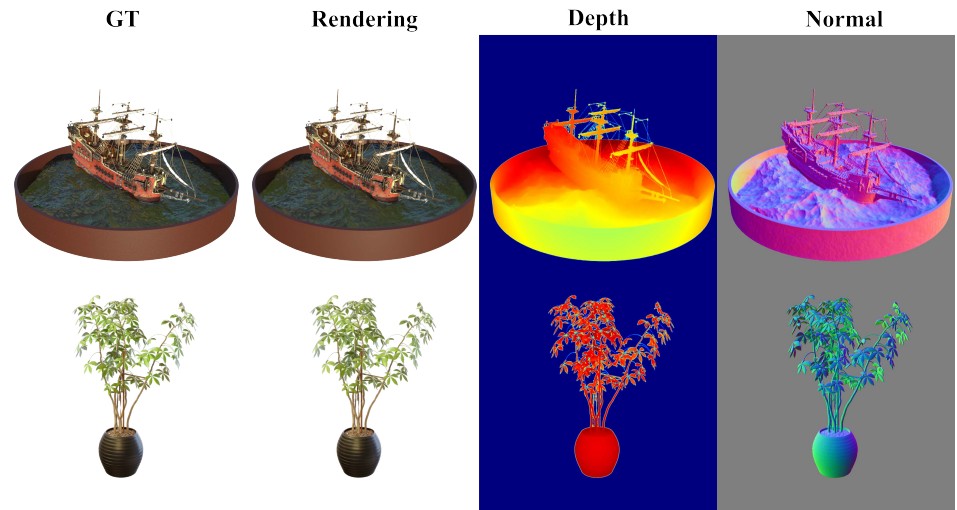

Figure 17: Additional reconstruction results on NeRF synthetic dataset with thin structures.

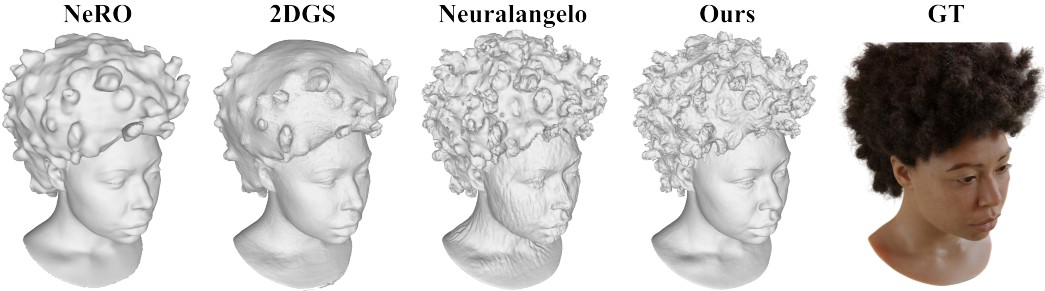

Figure 18: Comparison on the fuzzy object with previous surface reconstruction methods. AniSDF can reconstruct more accurate geometry of fuzzy object than other methods.

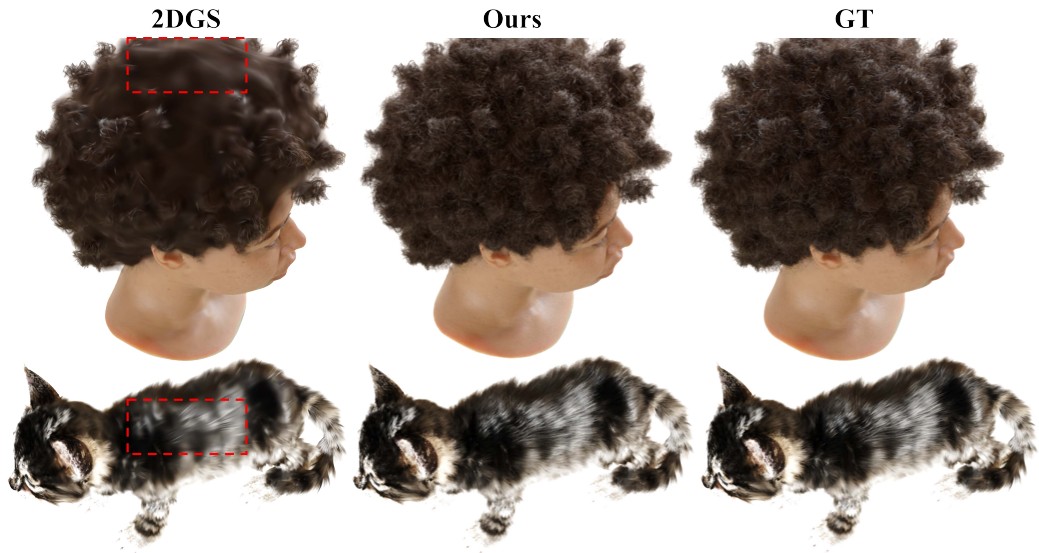

Figure 19: Rendering comparison on fuzzy object with 2DGS. Our method achieves better results for rendering hair and fur than 2DGS.

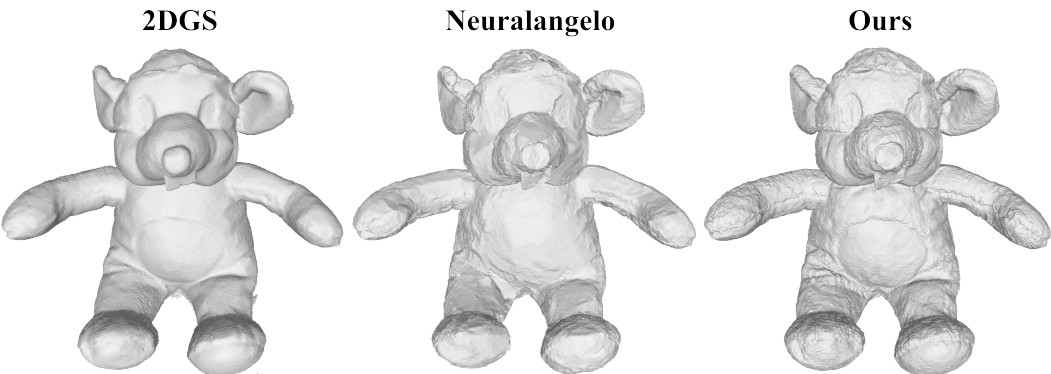

Figure 20: Comparison on DTU dataset. We demonstrate that AniSDF can reconstruct more detailed geometry than 2DGS and Neuralangelo.

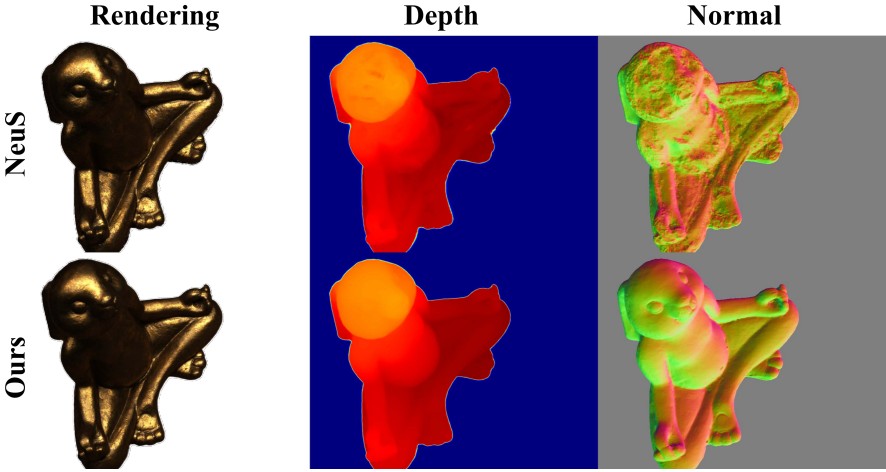

Figure 21: Comparison on DTU reflective dataset. Our method can reconstruct accurate surface for reflective objects.

### A.5 Possible Application

**Relighting.** AniSDF can provide accurate geometry for downstream application like inverse rendering and relighting. SUch tasks require accurate geometry for further materials estimation and relighting. We showcase the relighting results using the mesh generated by our method in Fig. 22.

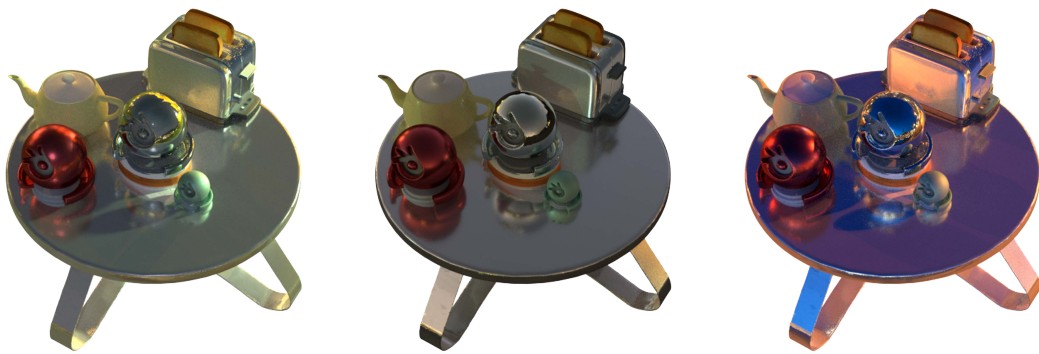

Figure 22: Relighting application demonstration. We showcase the relighing result based on our reconstructed geometry.

**Computer Graphics Animation.** We highlight that we can obtain accurate geometry for objects with thin structures and even candles in Fig. 23. Though the fire is not modeled by mesh in our physical world, we can utilize the accurate mesh of the candles for animation and render the animated results.

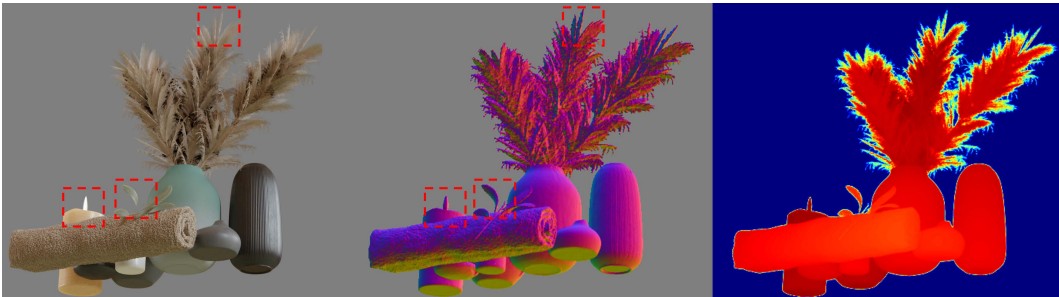

Figure 23: Animation application demonstration. Our method can reconstruct the mesh of candles that can be further used in animation.

