# OpenReview forum: "AniSDF: Fused-Granularity Neural Surfaces with Anisotropic Encoding for High-Fidelity 3D Reconstruction"
_ICLR.cc/2025/Conference — ICLR 2025 Poster_

### Official Review · Reviewer_nbo8 · 2024-10-29

**Soundness:** 2
**Presentation:** 2
**Contribution:** 2
**Rating:** 6
**Confidence:** 4

**Summary:**

The AniSDF paper introduces an innovative approach to high-fidelity surface reconstruction and photo-realistic rendering from multi-view images. This is achieved through a synergistic geometry network and appearance network, which together enable high-quality 3D reconstruction. Additionally, the authors propose a fused-granularity neural surface that aims to balance overall structural integrity with fine detail preservation.

**Strengths:**

Pros:
The paper is well-written and generally easy to follow.
Experimental results demonstrate incremental improvements in PSNR, which support the proposed approach.

**Weaknesses:**

Cons:
The fused-granularity neural surface structure may lack novelty, as it essentially uses two parallel structures with different resolutions. It seems likely that resolution choices could impact the final reconstruction quality. Including experiments that vary resolution settings would clarify their effect.


Despite claims of high-quality mesh reconstruction, Chamfer Distance results reveal performance gaps on certain objects (e.g., "Chair" and "Mic" categories) compared to methods like Neus and NeRO. Explaining these discrepancies would help elucidate the limitations.

**Questions:**

It’s unclear whether the larger network or the fused-granularity neural surface structure is responsible. What would happen if we set both the fine and coarse grids to the same resolution, either that of the coarse grid or that of the fine grid?

I raise my ratings. Good Luck.

---

> ### Author Response · Authors · 2024-11-18
> **Response to Reviewer nbo8**
>
> Thank you for your comments and please see our response to the feedbacks below.
>
> **Response to W1**: AniSDF achieves high-quality geometry reconstruction by fused-granularity structure and blended radiance field instead of solely increasing the maximum hashgrid resolution. Compared to Neuralangelo which set 4096 as the maximum resolution and $2^{22}$ as the maximum hash entry, we use hash-grid resolution from 32 to 2048 and entry as $2^{19}$, a lower resolution but a better performance. Our choice of 2048 hash-grid resolution mainly follows previous work Instant-NSR [1].
>
> To solve your concern about different resolution, we also conduct an ablation study of our resolution choice. In Fig.7, we freeze the minimal resolution as 32 and vary the maximum resolution from 1024 to 4096. It can be seen that by using 1024 as the maximum resolution, the model can not reconstruct some geometric details like the net of the sails. On the other hand, using 4096 as the maximum resolution can achieve slightly better results but requires larger memory consumption (32GB). Therefore, we use 2048 as the default resolution in our model (24GB).
>
> **Response to W2**: Since most methods tend to learn a solid geometry for those objects with hollow parts, e.g. Mic., Chamfer Distance is not always accurate to reflect the geometry quality. Due to its points sampling process on the mesh, it tends to produce better quantitative results for a smooth surface for these objects. As shown in Fig.10, our reconstructed geometry displays better visual effect than NeRO but gets a worse quantitative result. We also conduct an experiment by adjusting the hyper-parameters to produce a smoother surface and achieve the best metric.
>
> **Response to Q1**: We extend our ablation study (Fig.6) of the fused-granularity neural surfaces in Fig.8. Results with two fine branches with the same resolution setting produce a noisy surface and require larger memory consumption (30GB), while those with two coarse branches fail to reconstruct geometric details like the net of the sails. In contrast, the fused-granularity structure make the most of both branches and reconstruct the best results with moderate memory consumption (24GB).
>
> [1] Fuqiang Zhao, Yuheng Jiang, Kaixin Yao, Jiakai Zhang, Liao Wang, Haizhao Dai, Yuhui Zhong, Yingliang Zhang
> Minye Wu, Lan Xu, Jingyi Yu. Human Performance Modeling and Rendering via Neural Animated Mesh. In SIGGRAPH Asia, 2022.

---

> > ### Author Response · Authors · 2024-11-25
> > **Response to Reviewer nbo8**
> >
> > We are grateful for your great efforts in reviewing our paper. Your constructive feedbacks and valuable comments have significantly contributed to the improvement of our work. Since deadline of the discussion period is approaching, please, let us know if you have any additional concerns. We sincerely hope that our response will be considered during your assessment, and we can further address any clarifications or remaining issues.
> >
> > We would like to once again express our appreciation for the time and efforts you have dedicated to reviewing our paper.

---

> > > ### Comment · Reviewer_nbo8 · 2024-11-30
> > >
> > > Thank you for the response! I understand that using two coarse branches fails to reconstruct geometric details, but could you explain why using two fine branches leads to noisy surfaces? Additionally, do you have any suggestions on how to determine the resolutions for the fine and coarse branches, respectively? Are these resolutions consistent across different models?

---

> > > > ### Author Response · Authors · 2024-11-30
> > > >
> > > > Thank you for the constructive feedback!
> > > >
> > > > Using two fine branches leads to noisy surfaces due to the inaccurate learning of the appearance network. This is similar to only using high-resolution hashgrid in other single-branch neural SDF methods, as mentioned by several works like Neuralangelo and NeuS2. Specifically, at the early training stage, before the appearance network disambiguates appearance, the fine-grid can easily misinterprets specularity as redundant volumes. This kind of misinterpretation of appearance would then reflect to the geometry reconstruction, leading to fuzzy surface or hollow surface. From the perspective of model learning, surface reconstruction needs to strike a balance between the geometry network and appearance network. That is to say, the geometry network requires the texture network to disambiguate the appearance while the texture network requires the geometry network to be capable of representing detailed surfaces. Scaling up the resolution at the early stage breaks such balance by making the geometry network much more powerful than the texture network at early stage, and this leads to the overfitting of the geometry network.
> > > >
> > > > In AniSDF, we initially determine the resolutions for the fine and coarse branches by setting the level of $m$ right in the middle of the start level and final level. We select the start level and final level to be the same as Instant-NSR and we use the same config for all the dataset we use in the experiments section. The reason we implement is that we want both branches to learn the geometry in a same pace. Since the coarse branch converges faster than the fine branch, we want the fine branch to capture the geometric details step by step building upon the overall shape learned by the coarse branch. Therefore, we make both branches ''step" in the same scale. We also conducted ablation experiments on the level of $m$ to demonstrate this observation as shown in Fig.9. Since we implement the geometry network using hashgrids, these resolutions can be consistent across hashgrid-based neural SDF methods and we would try to conduct experiments in other methods.

---

> > > > > ### Author Response · Authors · 2024-12-02
> > > > >
> > > > > Dear Reviewer nbo8,
> > > > >
> > > > > We sincerely thank you for the efforts and time during the rebuttal. Since the extended discussion period is approaching to the end, we are wondering if you have any additional concerns. Please don't hesitate to let us know and we are more than willing to provide detailed explanations. Thank you again!
> > > > >
> > > > > *Best wishes*,
> > > > >
> > > > > *AniSDF Authors*

---

> ### Author Response · Authors · 2024-12-04
>
> We sincerely thank you for the constructive feedbacks and raising the final rating.

---

### Official Review · Reviewer_oJzK · 2024-11-02

**Soundness:** 3
**Presentation:** 3
**Contribution:** 2
**Rating:** 6
**Confidence:** 5

**Summary:**

The paper presents AniSDF, a novel SDF-based method for high-fidelity 3D reconstruction that incorporates fused-granularity neural surfaces and anisotropic spherical Gaussian (ASG) encoding. AniSDF aims to achieve accurate geometry reconstruction and photo-realistic rendering by addressing challenges in neural radiance fields, such as geometry-appearance trade-offs. The approach uses parallel fused-granularity neural surfaces to balance coarse and fine details, and blended radiance fields with ASG encoding for modeling both diffuse and specular appearances. Extensive experiments demonstrate AniSDF's superiority in both geometry reconstruction and novel-view synthesis over prior methods.

**Strengths:**

1.	Innovative Approach: The use of fused-granularity neural surfaces combined with ASG encoding for 3D reconstruction is novel and effective in balancing both coarse and fine details, leading to improved geometry and appearance quality.
2.	High Performance: Experimental results show significant improvements in rendering quality and geometry reconstruction, with better handling of reflective, luminous, and fuzzy objects compared to existing methods.

**Weaknesses:**

1.	Limited Real-Time Capability: AniSDF cannot perform real-time rendering, which limits its applicability in time-sensitive applications such as interactive graphics or augmented reality.
2.	Computation Cost: The use of multiple neural networks and high-resolution hash grids could be computationally expensive, which may hinder scalability.

**Questions:**

Questions:
1. I compared the chamfer distance metric of the Neuralangelo method reproduced on the DTU dataset in the paper, and there is a significant gap. The original paper reported an average of 0.61 (which surpasses your method), while the reproduced result in the paper is 1.07. Could the authors clarify the reasons for this discrepancy? Specifically, did you maintain the same hyperparameter settings as Neuralangelo? Please provide detailed information on your experimental setup.
2. Section 3.2 of the paper lists some issues related to coarse grid and fine grid training, but there are no corresponding experimental supports for these claims. Regarding the use of the coarse to fine method, you pointed out that thin structures may be discarded in the early training stages. Could you provide visualizations of surface reconstruction at different training stages, along with corresponding quantitative metrics, particularly for Neuralangelo and Neus2? This would help us assess the advantages and disadvantages in the reconstruction of detailed structures. I noticed your experiments in the ablation study, but they do not specify the experimental settings and only show the final results.

---

> ### Author Response · Authors · 2024-11-18
> **Response to Reviewer oJzK**
>
> Thank you for your comments and please see our response to the feedbacks below.
>
> **Response to Q1**: We reproduce the results using the hyperparameter settings in the official Neuralangelo code repository and use the official implementation when training. In fact, Neuralangelo requires a lot of parameter-tuning and it has not provided all optimal configs for each case. A similar result to our paper is also presented in Table.1 of UniSDF [1]. On the contrary, our method does not require complex parameter-tuning and achieve accurate results.
>
>
> **Response to Q2**: We have presented the visualization of the surface reconstruction process of Neuralangelo in the **rebuttal video**. Since geometry learning is supervised only by RGB color, as mentioned in 3.2, the hash grid would first learn the overall shape and try to learn thin structures in the later process. However, the plain coarse-to-fine hashgrid do not have spare power to learn the geometric details, thin structures are then discarded.
>
> To overcome this obstacles, increasing the maximum resolution and maximum hashgrid feature channel can lead to better results as proved by Neuralangelo. Our method on the other hand solve the problem without increasing these parameters and utilize the fused-granularity structure with the blended radiance field to provide an alternative solution that requires less computational resources than Neuralangelo with better results and faster training.
>
> **Response to Weakness**: Real-time rendering and computation cost are indeed two major weaknesses of AniSDF and we have already mentioned them in the Limitations section.
>
> [1] Fangjinhua Wang, Marie-Julie Rakotosaona, Michael Niemeyer, Richard Szeliski, Marc Pollefeys, Federico Tombari. UniSDF: Unifying Neural Representations for High-Fidelity 3D Reconstruction of Complex Scenes with Reflections. In NeurIPS, 2024.

---

> > ### Author Response · Authors · 2024-11-25
> > **Response to Reviewer oJzK**
> >
> > We are grateful for your great efforts in reviewing our paper. Your constructive feedbacks and valuable comments have significantly contributed to the improvement of our work. Since deadline of the discussion period is approaching, please, let us know if you have any additional concerns. We sincerely hope that our response will be considered during your assessment, and we can further address any clarifications or remaining issues.
> >
> > We would like to once again express our appreciation for the time and efforts you have dedicated to reviewing our paper.

---

> > > ### Comment · Reviewer_oJzK · 2024-11-25
> > >
> > > Thank you for your response and all of my concerns have been addressed. I maintain my original voting score.

---

> > > > ### Author Response · Authors · 2024-11-25
> > > >
> > > > We sincerely thank the reviewer for the constructive feedback and the exceptional efforts devoted to reviewing our paper!

---

### Official Review · Reviewer_7ZeX · 2024-11-04

**Soundness:** 3
**Presentation:** 3
**Contribution:** 3
**Rating:** 8
**Confidence:** 4

**Summary:**

This paper introduces AniSDF, an approach to enhance the quality of SDF-based methods in geometry reconstruction and novel-view synthesis tasks, enabling the physically-based rendering ability. Firstly, AniSDF uses the parallel branch structure of coarse hash-grids and fine hash-grids, replacing the former sequential coarse-to-fine training strategy, to learn a fused-granularity neural surface to improve the quality of SDF. Secondly, AniSDF uses Anisotropic Spherical Gaussian Encoding to learn blended radiance fields with a physics-based rendering, disambiguating the reflective appearance.

**Strengths:**

Originality:
The paper explores the potential of parallel using coarse and fine hash-grid to replace the general sequential coarse-to-fine structure, demonstrating the effects of experiments. Besides, this paper combines SDF learning with blended radiance field learning with anisotropic spherical Gaussian encoding to distinguish material information.

Quality:
The quality of the paper is good, evidenced by detailed experiments and comprehensive comparisons with state-of-the-art methods.

Clarity:
The paper is well-structured and organized.

Significance:
The good geometry that disambiguates the reflective appearance is helpful in 3D reconstruction. The possible relighting application makes this research meaningful.

**Weaknesses:**

1. Reference is insufficient: From Lines 130 to 135, the Sec. 2.1 is related to the attempts to improve the reconstructed geometry of Gaussians. Since 2DGS is used to compare, it is no reason to cite some papers focused on doing similar jobs: improving the surface reconstruction of 3DGS, like $\cite{guedon2023sugar, lyu20243dgsr, chen2023neusg}$.

2. The ablation study of the fused-granularity neural surface is not enough. The ablation study shows the comparison with the sequential coarse-to-fine method, but the technique with only coarse hash-grid and only fine hash-grid should also be demonstrated to prove the observations shown at the beginning of Sec.3.2. It could be better if the training time comparison is also shown in this part.

@article{guedon2023sugar,
  title={SuGaR: Surface-Aligned Gaussian Splatting for Efficient 3D Mesh Reconstruction and High-Quality Mesh Rendering},
  author={Gu{\'e}don, Antoine and Lepetit, Vincent},
  journal={CVPR},
  year={2024}
}
@article{lyu20243dgsr,
  title   = {3DGSR: Implicit Surface Reconstruction with 3D Gaussian Splatting},
  author  = {Xiaoyang Lyu and Yang-Tian Sun and Yi-Hua Huang and Xiuzhe Wu and Ziyi Yang and Yilun Chen and Jiangmiao Pang and Xiaojuan Qi},
  year    = {2024},
  journal = {arXiv preprint arXiv: 2404.00409}
}
@article{chen2023neusg,
  title   = {NeuSG: Neural Implicit Surface Reconstruction with 3D Gaussian Splatting Guidance},
  author  = {Hanlin Chen and Chen Li and Gim Hee Lee},
  year    = {2023},
  journal = {arXiv preprint arXiv: 2312.00846}
}

**Questions:**

1. Modify the typo in Line 093 (‘Oue’) to ‘Our.’

2. The physical-based rendering method via ASG is similar to $\cite{yang2024spec}$. Is this work also inspired by similar works using ASG to learn the specular representation in 3DGS?

3. The blended radiance fields with ASG encoding are composed of $c_{view}$ and $c_{ref}$ though a learnable weight. According to Eq. 4, the light field is modeled by $c_d$ and $c_s$, diffuse color and specular color. So, can $c_{view}$ be regarded as purely diffuse and $c_{ref}$ as the pure composition of specular? If so, can the radiance fields be considered purely diffuse when $\omega$ is 1? If not, can this work disentangle the light field to only diffuse or specular field?

4. Refer to $\cite{han2023multiscale}$, they control the final color by adding the scale to the color calculated from ASG when retaining the diffuse color term calculated from the first three orders of SH. So, what motivates adding weight to diffuse and specular in this work? What are the differences between your light field calculation in the blended radiance fields with $\cite{han2023multiscale}$?

@article{han2023multiscale,
  title     = {Multiscale Tensor Decomposition and Rendering Equation Encoding for View Synthesis},
  author    = {Kang Han and Weikang Xiang},
  journal   = {Computer Vision and Pattern Recognition},
  year      = {2023},
  doi       = {10.1109/CVPR52729.2023.00412},
  bibSource = {Semantic Scholar https://www.semanticscholar.org/paper/aa41843888fffada6335b6c5cdbcd2d4bb5cf9da}
}

@article{yang2024spec,
  title={Spec-gaussian: Anisotropic view-dependent appearance for 3d gaussian splatting},
  author={Yang, Ziyi and Gao, Xinyu and Sun, Yangtian and Huang, Yihua and Lyu, Xiaoyang and Zhou, Wen and Jiao, Shaohui and Qi, Xiaojuan and Jin, Xiaogang},
  journal={arXiv preprint arXiv:2402.15870},
  year={2024}
}

**Details Of Ethics Concerns:**

The reviewer would like to raise awareness of possible breach of double blind review rules.

The reviewer found a twitter page:

https://x.com/zhenjun_zhao/status/1842119223646302292

This page introduces their paper with authors names explicitely shown.

---

> ### Author Response · Authors · 2024-11-18
> **Response to Reviewer 7ZeX**
>
> Thank you for your comments and please see our response to the feedbacks below.
>
> **Response to W1**: Thanks for the wonderful suggestion of the references, we have added these references into our latest version.
>
> **Response to W2**: Thanks for your advice. We have conducted an ablation study to compare the performance of both branches in coarse (coarse and coarse) and both branches in fine (fine and fine) with that of our current setting (coarse and fine). As shown in Fig.8, the results of two fine branches are noisy and fuzzy, particularly on some smooth surfaces, while two coarse branches cannot reconstruct thin structures, e.g., net of the sails. This is also consistent with our statement in 3.2 of the original paper. We also compare the overall training time of the same iterations, setting both branches to coarse requires 1h40min and both branches to fine requires 3h20min. The fused-granularity structure (coarse+fine) requires about 2h20min to converge.
>
> **Response to Q1**: Thanks for the careful checking, we have modified the typo.
>
> **Response to Q2**: Spec-gaussian is another great concurrent work incorporating ASG to learn specularity. Our work is originally motivated by NRFF [1]. We are motivated by the ASG’s ability to capture the anisotropy. In our work, we then use ASG encoding to disambiguate view-dependent effects such as specularity for better surface reconstruction. It enables learning accurate geometry, while the motivation of spec-gaussian falling in the scope of obtaining better rendering. We have added the reference of spec-gaussian in our manuscript.
>
> **Response to Q3**: Our method mainly focus on reconstruction instead of material decomposition. The reason we use $c_{view}$ and $c_{ref}$ is to disambiguate the geometry from the reflective appearance. Since our method can be considered as a prior-free method, we do not introduce such materials priors in our model and the process of decomposing the scene is fully done by exploiting the power of the texture network with ASG encoding. So $c_{view}$ does not equal to the well-known diffuse color and $c_{ref}$ does not equal to the well-known specular color.
>
> However, $c_{view}$ can be viewed as the approximation of diffuse and $c_{ref}$ as the approximation of specular. Our work then can be considered as disentangling the light field to only the "diffuse" field or "specular" field. For real-world scenes, one scene is never fully diffuse or specular, our method obeys such rules and try to split the into $c_{view}$ and $c_{ref}$. For synthetic scenes, one scene can be fully diffuse of specular, then the weight would be leaning to 1 and obtain a value like 0.999.
>
> **Response to Q4**: NRFF [1] incorporates (diffuse + tint*specular) when rendering, and it utilizes an additional channel to estimate the value of tint. In the experimental setting of NRFF [1], it utilizes a neural network with more layers and larger hidden layers dimension for optimization. However, since we want to reconstruct high-quality surface in a comparable training with moderate computational resources, our method have smaller neural network. Learning the value of tint may cause additional pressure on the texture network. Therefore, we split the learning of this value by introducing a small network that optimizes the weight, where the radiance fields continuously determine and use the most adapted parametrization for each surface area.
>
> [1] Kang Han, Wei Xiang. Multiscale tensor decomposition and rendering equation encoding for view synthesis. In CVPR, 2023.

---

> > ### Author Response · Authors · 2024-11-25
> > **Response to Reviewer 7ZeX**
> >
> > We are grateful for your great efforts in reviewing our paper. Your constructive feedbacks and valuable comments have significantly contributed to the improvement of our work. Since deadline of the discussion period is approaching, please, let us know if you have any additional concerns. We sincerely hope that our response will be considered during your assessment, and we can further address any clarifications or remaining issues.
> >
> > We would like to once again express our appreciation for the time and efforts you have dedicated to reviewing our paper.

---

> > > ### Author Response · Authors · 2024-12-02
> > >
> > > Dear Reviewer 7ZeX,
> > >
> > > We sincerely thank you for the efforts and time during the rebuttal. Since the extended discussion period is approaching to the end, we are wondering if you have any additional concerns. Please don't hesitate to let us know and we are more than willing to provide detailed explanations. Thank you again!
> > >
> > > *Best wishes*,
> > >
> > > *AniSDF Authors*

---

### Official Review · Reviewer_ArLb · 2024-11-05

**Soundness:** 3
**Presentation:** 3
**Contribution:** 2
**Rating:** 6
**Confidence:** 4

**Summary:**

This paper provides a fused-granularity neural surfaces with physics-based anisotropic
spherical Gaussian encoding for high-fidelity 3D reconstruction. The authors show state-of-the-art novel-view rendering and geometry reconstruction results on several datasets, including NeRF-Synthetic, Shiny Blender, and DTU datasets. The proposed method shows very convincing reconstruction of challenging specular and furry objects.

**Strengths:**

1. This paper is well-written and easy to follow.
2. Qualitative and quantitative results of the proposed method seems very strong, beating prior baselines like NeuS, RefNeRF, RefNeuS. Reconstructed meshes look clean with high-quality surface normals.
3. The authors validate the proposed method on both synthetic datasets (Nerf-synthetic and Shiny-blender), and real datasets (DTU), and it's nice to see improvements.

**Weaknesses:**

1. To further convince me about the method's performance on reconstructing specularity, I would need to see the view synthesis videos, as opposed to the static frames shown in the paper and project page. Unfortunately, I could not find such videos (except the relighting ones).

2. For the proposed blended radiance field (Eq. 12), I think it would be great to provide some visualizations of the individual components: w, c_view, c_ref, to better understand what each learnt components look like.

3. I'm unsure why the fused-granularity hash grids actually works better than a plain multi-resolution hash grids. It seems to me that the major difference of it from a plain one is the additional handcrafted equation (6) that says the final SDF is an addition of coarse and fine SDF. In the plain multi-resolution hash grid, the final SDF is predicted by a MLP from concatenated multi-resolution features. This could benefit some justification.

**Questions:**

1. For Eq. 10, why would both c_view and c_ref depend on view directions? Would it encourage better diffuse-specular separation if one makes c_view view-independent?

---

> ### Author Response · Authors · 2024-11-18
> **Response to Reviewer ArLb**
>
> Thank you for your comments and please see our response to the feedbacks below.
>
> **Response to W1**: We have added two novel view synthesis results in the **rebuttal video** from the shiny blender datasets, it can be seen that we can reconstruct specularity and normal well.
>
> **Response to W2**: We have provided the visualization of $c_{view}$ and $c_{ref}$ for reference. As shown in Fig.11, our methods can well separate reflectance from base color.
>
> **Response to W3**: The coarse grid focuses more on the overall structures and tends to discard the details in the early stage. Compared to the overall structure of ship, the net of the sails occupies a very small proportion in the renderings, and thus setting both branches to coarse fail to reconstruct the net as shown in Fig.8. In contrast, the fine grid pays more attention on the geometric details and may overlooks the global shape, so setting both branches to fine may induce noise on smooth surface.
>
> As shown in the **rebuttal video**, sometimes thin structures that are already discarded in the early stage cannot be restored in the following stage. Of course in the context of plain multi-resolution hashgrids, this can be avoided by increasing the resolution or the block size infinitely to utilize the learning power, which would lead to much higher computational recourses and longer training time like Neuralangelo. Our method on the other hand use fuse-granularity as an alternative to avoid such scaling problems.
>
>  **Response to Q1**: Thanks for the wonderful suggestion. Since AniSDF focuses on surface reconstruction, decomposing the diffuse and specular color is not the primary task. The goal of our task is to reduce the ambiguity brought by the appearance when reconstructing geometry from multi-view images. In this case, the disentanglement process can be viewed as a classifying problem, that is, we want to guide the texture network learn whether the specific surface area belongs to “diffuse” or “specular”. Following the reparametrization introduced by RefNeRF, the reflective direction works well as a guidance of specularity. Therefore, we choose the other classification guidance to be the input view direction d, as the differences between two directions are already sufficient for “diffuse”-“specular” separation.
>
> Setting $c_{view}$ as view-independent is an insightful idea. We have conducted experiment by omitting the $d$ term in the MLP input to make $c_{view}$ view-independent. As shown in Fig.13, the decomposition has subtle differences and with $d$ as an additional input, the model can achieve slightly better PSNR results (26.98 vs. 26.84).

---

> > ### Author Response · Authors · 2024-11-25
> > **Response to Reviewer ArLb**
> >
> > We are grateful for your great efforts in reviewing our paper. Your constructive feedbacks and valuable comments have significantly contributed to the improvement of our work. Since deadline of the discussion period is approaching, please, let us know if you have any additional concerns. We sincerely hope that our response will be considered during your assessment, and we can further address any clarifications or remaining issues.
> >
> > We would like to once again express our appreciation for the time and efforts you have dedicated to reviewing our paper.

---

> > > ### Author Response · Authors · 2024-12-02
> > >
> > > Dear Reviewer ArLb,
> > >
> > > We sincerely thank you for the efforts and time during the rebuttal. Since the extended discussion period is approaching to the end, we are wondering if you have any additional concerns. Please don't hesitate to let us know and we are more than willing to provide detailed explanations. Thank you again!
> > >
> > > *Best wishes*,
> > >
> > > *AniSDF Authors*

---

### Official Review · Reviewer_JMuD · 2024-11-07

**Soundness:** 2
**Presentation:** 3
**Contribution:** 3
**Rating:** 6
**Confidence:** 4

**Summary:**

The paper presents a high-quality 3D surface reconstruction method- AniSDF, which learns fused-granularity neural surfaces with physics-based encoding. The authors propose fused multi-resolution grids for geometry modeling, and adopt Anisotropic Gaussians for appearance modeling. With these designs, AniSDF can reconstruct objects with complex structures and produce high-quality renderings on benchmarked datasets.

**Strengths:**

1. The paper is well written and easy to follow. The idea is clean and the pipeline do not introduce additional hyper-parameter tuning and selection compared to some other recent methods for neural surface reconstruction / rendering.
2. The idea to fuse multi-resolution grids for detailed surface reconstruction is novel. AniSDF’s fused-granularity structure balances high- and low-resolution information to improve convergence and accuracy. This approach allows for a more adaptive reconstruction that captures both overall structure and fine details, which is validated by their good geometry quality (chamfer) in the experiments.
3. The use of ASG encoding in appearance modeling seems to be effective and handles specular reflections very well.

**Weaknesses:**

1. Experiments for reflective surfaces mainly come from synthetic data, it would be helpful to understand the model’s ability if we could see results of more real-world reflective surface data, such as trucks from Tanks and Templates, sedans from Refnerf, The Glossy-Real dataset from Nero.
2. The appearance modeling involves blending view-based and reflection-based radiance fields, however, the method's ability to decompose base color and reflection color is unknown.  It would be better if the author could add a visualization of view-based color, reflection-based color, and the blending weight.
3. Some details of the methods are missing. The derivation of normal and the method for mesh extraction are not discussed.
4. The reason behind the choice of grid level `m` and `l`  is not clear. It would be clearer if an ablation study about grid resolution were added.

**Questions:**

1. Add real-world experiments and more benchmark datasets (with larger scene scale).
2. Add visualizations to the decomposed appearance throughout the figures in the paper.
3. Add missing details in the method section.
4. Add some comparison on the train - test efficiency and memory footprint.
4. Revised the discussions and ablation studies as suggested in the weaknesses.

---

> ### Author Response · Authors · 2024-11-18
> **Response to Reviewer JMuD**
>
> Thank you for your comments and please see our response to the feedbacks below.
>
> **Response to W1(Q1)**: Larger-scene scale experiments can be shown in the Appendix A.2 Unbounded Scenes section where we use MipNeRF360 as the dataset. To further demonstrate AniSDF’s ability to reconstruct reflective surfaces, we have also provided the reconstruction results on NeRO (Coral) and RefNeRF-Real (Sedan and Gardenspheres) as results in Fig.12.
>
> **Response to W2(Q2)**: We have provided the visualization of $c_{view}$ and $c_{ref}$ for reference. As shown in Fig.11, our method achieves great results on separating reflectance from base color.
>
> **Response to W3(Q3)**: We derive our normal as the gradient of the signed distance field: $\mathbf{n}=\nabla d(\mathbf{x}) /\|\nabla d(\mathbf{x})\|$ like most of the other SDF-based methods. As for the mesh extraction, we use marching cubes as the techniques following the NeuS pipeline. We would add these details later in the methods section of the manuscript.
>
> **Response to W4(Q5)**: The choice of $l$ is to follow the implementation of Instant-NSR [1], and we choose $m=10$ to be right in the middle of the beginning level 4 and the ending level $l=16$, where we want the parallel structures each contain the same levels.
>
> Though this is an empirical hyper parameter setting, we also conduct an ablation experiment on the grid resolution. As shown in Fig.9, by setting $m=9$, some geometric details are omitted when learning and by setting $m=11$, some smooth surface can not be reconstructed. This can be interpreted as another demonstration of the observations in 3.2.
>
> **Response to Q4**:
> |    Method    | Training Time  | Memory Consumption  |
> |:------------:|:--------------:|:-------------------:|
> |     NeuS     |   8-10 hours   |         20GB        |
> |     NeRO     |   16-18 hours  |         20GB        |
> | Neuralangelo |   20-22 hours  |         32GB        |
> |     Ours     |    2-3 hours   |         23GB        |
>
> We have added the comparison of the surface reconstructed method on the training efficiency and memory footprint. It can be seen that with comparable memory consumption, our method requires the leasting training time while obtaining the best results.
>
>
> [1] Fuqiang Zhao, Yuheng Jiang, Kaixin Yao, Jiakai Zhang, Liao Wang, Haizhao Dai, Yuhui Zhong, Yingliang Zhang
> Minye Wu, Lan Xu, Jingyi Yu. Human Performance Modeling and Rendering via Neural Animated Mesh. In SIGGRAPH Asia, 2022.

---

> > ### Author Response · Authors · 2024-11-25
> > **Response to Reviewer JMuD**
> >
> > We are grateful for your great efforts in reviewing our paper. Your constructive feedbacks and valuable comments have significantly contributed to the improvement of our work. Since deadline of the discussion period is approaching, please, let us know if you have any additional concerns. We sincerely hope that our response will be considered during your assessment, and we can further address any clarifications or remaining issues.
> >
> > We would like to once again express our appreciation for the time and efforts you have dedicated to reviewing our paper.

---

> > > ### Author Response · Authors · 2024-12-02
> > >
> > > Dear Reviewer JMuD,
> > >
> > > We sincerely thank you for the efforts and time during the rebuttal. Since the extended discussion period is approaching to the end, we are wondering if you have any additional concerns. Please don't hesitate to let us know and we are more than willing to provide detailed explanations. Thank you again!
> > >
> > > *Best wishes*,
> > >
> > > *AniSDF Authors*

---

> ### Comment · Reviewer_JMuD · 2024-12-03
>
> I appreciate the authors for the experiments and visualizations added to the manuscript. Part of my concern is addressed, and I have raised my final rating.

---

> > ### Author Response · Authors · 2024-12-03
> >
> > We sincerely thank the reviewer for the constructive feedback and thank you for raising the final rating. We would continue to improving the final paper.

---

### Author Response · Authors · 2024-11-18
**General Response to All Reviewers**

We want to thank all the reviewers for their time and valuable comments.

Here we summarize the changes we have made to our manuscript:

1. We provide additional experiments results in the **Appendix A.1 Additional Results for Rebuttal**, the major components are:
- Fused-Granularity Neural Surfaces Ablation. We demonstrate the effectiveness of our fused-granularity structure by showcasing the experiments on maximum hashgrid resolution and different granularity structure.
- Blended Radiance Fields Additional Results. We demonstrate several results including the visualization of $c_{view}$ and $c_{ref}$  and real-world reflective surface reconstruction results.
2. We showcase a **rebuttal video** to demonstrate additional results in the supplementary files.
3. We include some recommended references in the Related Work section.
4. We modifiy some typos noticed by some reviewers.

---

### Author Response · Authors · 2024-11-28

Dear Reviewers, ACs, and PCs,

We sincerely thank all the reviewers for their valuable and constructive feedback, as well as for dedicating their efforts and time to reviewing our paper. Based on the suggestions provided during the discussion period, we have revised our paper to address the reviewers' key concerns. We hope the revision provides the reviewers, ACs, and PCs with a clearer understanding of the progress. Please let us know if you have any additional concerns and we are more than willing to provide detailed explanations for any further concerns raised by the reviewer.


Once again we deeply appreciate the efforts of the reviewers, ACs, and PCs during the discussion period.


*Best regards*,

*AniSDF Authors*

---

### Meta-Review · Area_Chair_V2HK · 2024-12-21

**Metareview:**

The paper proposes a mechanism for high-fidelity 3D reconstruction, showing state-of-the-art results on reconstruction and novel-view synthesis, providing particular gains in view-dependent effects and thin structures, e.g., hair. Reviews are unanimously positive, albeit not all very strongly.

Reviewers comment on the clarity of manuscript, effectiveness of proposed approach quality of results on classically challenging cases, and comprehensive evaluations.

A number of weaknesses are also highlighted, and authors are encouraged to make updates to the manuscript before the final camera-ready: added ablation studies and other experiments/comparisons, added visualizations/videos, more fleshed out limitations in paper prose, to name a few.

**Additional Comments On Reviewer Discussion:**

Reviews began as closer to borderline and were raised slightly as the authors addressed reviewer concerns about missing citations and other minor issues in the paper. There remain a number of smaller fixes to be made, but the reviewers seem to have been left content about the current state of the paper.

---

### Decision · Program_Chairs · 2025-01-22

Accept (Poster)